# Towards a Collaborative Memory for Agentic Workflow: Breaking the Prefix Barrier with Segment-Level KV Cache Sharing

## Abstract

In LLMs-based multi-agent systems, the Key-Value (KV) cache serves as a critical carrier of agents' working memory, and its efficient reuse is paramount for enhancing the service throughput and inference efficiency. However, prevailing KV cache reuse methods rely heavily on a rigid prefix matching mechanism, which mandates exact equivalence between the query request and the cached prefix. This inflexible matching scheme struggles to accommodate the highly heterogeneous instruction prompt template in multi-agent environments, thereby severely constraining the overall system throughput. To overcome these limitations, this paper introduces a novel collaborative memory approach, underpinned by a Segment-Level KV Cache Sharing mechanism. This method decomposes the cache unit into fine-grained semantic segments, enabling agents to dynamically reuse KV cache segments generated by any other agent at arbitrary positions, without relying on sequential prefix consistency. Our approach not only significantly boosts the inference efficiency of LLMs in agentic workflows but also achieves genuine working memory sharing and collaboration, thereby enhancing cooperative capabilities among agents. Our implementation is built upon the vLLM framework and leverages the PageAttention mechanism. Extensive experimental results demonstrate that the proposed method markedly reduces redundant computation, increases system throughput, and even improves the performance of agentic workflow on benchmark tests through effective working memory sharing.

## 1 Introduction

LLM-based agents have demonstrated strong performance across a wide range of domains. Multi-agent workflows have further shown significantly superior general reasoning capabilities, achieving enhanced performance on various benchmark tasks (Li et al., 2023; Hong et al., 2024; Wu et al., 2024; Parmar et al., 2025). Memory mechanism, as one of the most important components of agent systems, has been widely explored to enhance reasoning (Long et al., 2025), planning (Fang et al., 2025), and personalization (Cai et al., 2025). In multi-agent systems (MAS), memory plays an even more critical role by not only supporting individual agents' cognitive processes but also facilitating coordination and collective intelligence among agents (Hong et al., 2024; Yan et al., 2025).

The representation of memory for LLM can be broadly categorized into three types (Wu et al., 2025; Zhang et al., 2025b): *textual memory* that stores historical interactions or summaries in natural language, *knowledge-graph memory* that organizes structured knowledge with entities and relations, and *parametric memory* that stores information within model parameters or intermediate activations. KV cache, as one of the most important types of parametric memory (Wu et al., 2025), preserves critical short-term informational states and serves as the essential working memory. However, while extensive recent works (Chhikara et al., 2025; Edge et al., 2024; Zhang et al., 2025b) have explored textual memory, knowledge graphs, and LoRA-based parametric adaptations, few have investigated the sharing of KV cache, especially in multi-agent settings. This overlooks its significant potential: the KV cache inherently captures rich information and attention patterns that could be leveraged.

Efficient reuse of KV cache across agents would not only substantially improve inference efficiency but also enable dynamic, real-time working memory sharing, thereby enhancing collective intelli-

gence. On the one hand, it supports efficient inference by caching intermediate states and eliminating redundant computations. As the capabilities of large language models (LLMs) continue to improve, both model size and the computational expense of long-context inference have risen significantly. This issue is particularly acute in MAS, where frequent interactions among agents rapidly expand contextual data, drastically reducing per-agent prefill throughput and compromising overall system efficiency. Empirical observations (Shen et al., 2025; Pan et al., 2025) reveal that extended negotiation cycles often produce substantial redundant content that is repeatedly exchanged among agents. A canonical illustration is a project-plan artifact produced by a manager agent that is subsequently disseminated to, and re-processed by, dozens of downstream agents, resulting in prohibitive computational overhead. On the other hand, reusing KV cache enables sharing of reasoning information among agents, allowing them to leverage each other's cognitive processes, i.e., the intermediate representations and thought processes underlying decisions. This capability is essential in collaborative tasks requiring high contextual consistency and coordinated reasoning. For instance, in a software development MAS where multiple agent roles (e.g., architect, programmer, tester) are required to collaborate, sharing the working memory related to a module's design rationale or error analysis allows downstream agents to inherit context and continue reasoning without recapitulating earlier steps. By facilitating such shared cognitive workflows, working memory fosters deeper collaboration, reduces inconsistency, and elevates collective intelligence. Overall, effective working memory sharing not only reduces latency but also facilitates emergent cooperative behaviors, making it a cornerstone for building scalable and coherent multi-agent architectures.

Nevertheless, naive application of prefix caching (Kwon et al., 2023; Zheng et al., 2024) often fails to ameliorate this redundancy and share memory, since it requires exact query alignment for cache reuse, which becomes a significant bottleneck in MAS. The inherent heterogeneity of queries in MAS, such as diverse system prompts template among agents and tasks, makes exact prefix matches rare and hinder efficient cross-agent memory sharing. Some recent works attempt to accelerate the inference speed of agentic workflow by maximizing prefix hits in the prompt KV cache. KVFlow (Pan et al., 2025) reveals the inefficiency of the least recently used (LRU) eviction policy and introduces a workflow-aware eviction policy for history kv caches. Other works Liu et al. (2024a); Shen et al. (2025) are dedicated to adjust the order of queries to increase the cache hit rate. DroidSpeak (Liu et al., 2024b) focuses on the prefix-caching between two LLMs, and proposes to choose the critical layer groups to recompute and reuse KV cache for other layers. However, these works are still constrained by the prefix caching that have the exact same prefix content and overlook the KV cache reusing of a specific segment of content across agents.

To this end, we propose a segment-level KV cache sharing mechanism, which eliminates the requirement for strict prefix alignment and enables fine-grained, content-driven cache sharing across agents. Specifically, we treat each agent's output as a self-contained semantic segment and store its corresponding full KV cache. When another agent's query contains content that matches a previously generated segment, even if it is embedded within a longer or distinct context, the system efficiently retrieves and reuses the exact KV cache associated with that segment, bypassing the need for recomputation. To verify the feasibility of the proposed approach, we conducted a visual analysis experiment comparing the attention maps between tokens during the decoding phase with and without segment-level KV cache reuse. The results reveal that the two settings produce attention maps with highly similar patterns, providing visual evidence that the cached segments preserve coherent contextual relationships. As illustrated in Fig. 1 and discussed in detail in Sec. 3.1.2, this similarity confirms that the KV cache of a segment generated by one agent can be directly and effectively reused by other agents when the segment appears in a new query, even if their input prefixes differ.

Furthermore, we implement a high-performance prototype, dubbed CrossKV, atop the vLLM platform to effectively implement the proposed segment-level KV cache sharing mechanism. Built upon vLLM's paged attention technique, our method employs a `Memory Table` that records the logical hash identifiers of semantic segments that can be used to retrieve the physical blocks of their corresponding KV caches. Specifically, CrossKV consists of three coordinated stages: (1) Pre-Alias during Pre-Filling: The system first identifies reusable segments within the input, and then queries the `Memory Table` and vLLM's internal `Prefix Hash Table` to retrieve the physical blocks of these segments. Finally, the system assigns new logical hashes that are calculated based on the current context to these physical blocks, effectively creating aliases that enable efficient cross-context cache reuse; (2) Decoding: The decoding stage leverages vLLM's paged attention mechanism to perform inference efficiently, incorporating the KV cache without interruption to the token generation

process; (3) Post-Recording: After an agent completes its output, the system regards the response as a semantic segment, computes logical hashes, and records new entries in the `Memory Table` that associate these hashes with the segment, whose KV cache blocks can be efficiently queried to facilitate reuse in subsequent queries by other agents. Through this streamlined process, our approach demonstrates substantial improvements in multi-agent inference efficiency. Experimental results indicate that CrossKV achieves a system speedup of 4.6× for Time-To-First-Token (TTFT) over baseline methods while preserving output accuracy, and even exceed the baselines on multiple benchmarks due to efficient information sharing. Our contributions are summarized as follows:

**1) A Novel Segment-Level KV Cache Sharing Mechanism to Overcome Prefix Caching Limitation.** We identify the inherent limitation of prefix-caching paradigm in multi-agent collaboration and propose a novel segment-level KV cache sharing mechanism. It enables agents to dynamically reuse KV segments regardless of their position in the query, thereby overcoming prefix mismatch issues arising from the heterogeneity and dynamicity of instruction prompt templates for multi agents.

**2) An Efficient Prototype Enabling Cross-Agent Working Memory Sharing.** We implement a high-performance prototype, dubbed CrossKV, atop the vLLM inference engine, leveraging PageAttention for efficient KV cache management. Our design is model-agnostic allowing seamless integration with multiple LLMs, enabling rapid deployment as a service, facilitating flexible and efficient working memory sharing across diverse multi-agent workflows.

**3) Empirical Validation Demonstrating Both Efficiency and Collaborative Gains.** Extensive experiments demonstrate that our method significantly reduces redundant computation and improves the inference efficiency of multi-agent collaboration. Moreover, by enabling effective segment-level working memory sharing, it can enhance the performance of agentic workflows on multiple benchmarks. These results confirm that our approach is not merely a systematic acceleration technique, but also a new paradigm for strengthening collaborative intelligence among agents.

## 2 RELATED WORK

Agentic workflows leverage collaboration among multiple specialized agents to tackle complex tasks, demonstrating enhanced capabilities across diverse domains. Several studies focus on specific applications, such as software development (Qian et al., 2023; Hong et al., 2024), while others employ multi-agent debate to strengthen reasoning processes (Liang et al., 2024). General-purpose frameworks (Wu et al., 2024; Li et al., 2023) further allow users to design customized multi-agent interactions. A common challenge across these systems, however, is the significant repetition of content during inter-agent communication, leading to computational inefficiency.

**Agent Memory Mechanism.** Effective memory mechanisms is essential for transformer-based LLMs. Recent research has explored diverse memory architectures: Memory-R1 (Yan et al., 2025) uses reinforcement learning to enable autonomous memory management; REMI (Raman et al., 2025) builds personalized causal knowledge graphs for individualized reasoning; Nemori (Nan et al., 2025) introduces a cognitively-inspired structure that interconnects and layers memories by frequency of use. Domain-specific efforts, such as EndoAgent (Tang et al., 2025) in medical imaging, combine long- and short-term memory to improve diagnostic accuracy. Despite these advances, most works focus on long-term and episodic memory, leaving working memory, especially KV cache as a memory carrier, under-explored.

**KV Cache Sharing.** KV caching is widely adopted to accelerate LLM inference by storing intermediate key-value states and avoiding redundant computation. Conventional systems often rely on prefix-based reuse strategies (Kwon et al., 2023), where cached content is matched via exact prefix alignment. Recent efforts like CacheBlend (Yao et al., 2025) and KVShare (Yang et al., 2025) explore content-aware caching beyond strict prefix constraints for RAG system. However, few works have been proposed to address the dynamic, heterogeneous nature of multi-agent interactions, where flexible and efficient cross-agent KV cache sharing is critically needed.

## 3 THE PROPOSED CROSSKV

This section delineates the architecture and implementation details of our proposed collaborative memory framework for agentic workflow, dubbed CrossKV. Sec. 3.1 first introduces the core con-

cept of the segment-level KV cache sharing mechanism, explaining its design rationale and operational validity. Subsequently, Sec. 3.2 elaborates on the development of an efficient prototype built upon the vLLM framework, which materializes the envisioned cross-agent working memory sharing. Finally, Sec. 3.3 engages in a pertinent discussion about the impact of positional encoding, addressing a key technical challenge associated with KV cache reuse.

### 3.1 SEGMENT-LEVEL KV CACHE SHARING MECHANISM

#### 3.1.1 FROM PREFIX-CACHING TO SEGMENT-LEVEL KV CACHE IN AGENTIC WORKFLOW

The dominant approach for reusing KV caches, known as prefix caching, is predicated on a strict requirement: the initial tokens of an incoming query must perfectly match those of a previously computed sequence in order to leverage cached key-value states. Such a method will exhibit critical shortcomings in agentic workflows. The diversity among agents, arising from distinct system prompt templates, specialized roles, and individualized goals, will frequently result in different query prefixes. Even a subtle change can lead to failure of prefix caching. Moreover, the iterative and adaptive nature of multi-agent collaboration causes rapid divergence in dialogue or task context, further diminishing the likelihood of prefix overlap. As a result, opportunities for cache reuse are scarce, compelling agents to repeatedly recompute KV cache for semantically equivalent content. This redundancy not only constrains system throughput but also constitutes a fundamental scalability barrier in resource-intensive applications.

To address these inherent constraints, we introduce a segment-level KV cache sharing mechanism. Building upon theoretical insights into Transformer self-attention mechanisms, including their inherent sparsity (Farina et al., 2024; Lou et al., 2024; Ge et al., 2024), locality bias (Zhang et al., 2025a), and low-rank approximation properties (He et al., 2025), we posit that a continuous span of tokens with relatively independent semantics, namely a semantic segment, encodes locally concentrated and largely context-independent representations. Therefore, the KV cache of such a segment remains stable and reusable across different contextual settings. Formally, suppose $Q_A$ denotes the input query of Agent-A and $O_A = \text{LLM}([Q_A])$ is the output of Agent-A. Suppose Agent-B's input is $Q_B = [Q_{B1}, O_A, Q_{B2}]$. During the pre-filling phase for $Q_B$, the KV cache for the exact segment $O_A$, computed during Agent-A's decoding, can be reused.

The core principle of our method is to treat the continuous token sequence generated by an agent as a self-contained semantic unit whose KV cache captures essential reasoning steps and can be efficiently shared. By decoupling cache reuse from rigid prefix alignment, our approach enables flexible and efficient sharing of intermediate computations across agents, independent of variations in their initial instructions or broader context. This mechanism supports more adaptive agent instruction designs, allowing prompts to be placed either before or after reused segments while maintaining high cache hit rates and minimal performance degradation.

Supported by both sparsity property of attention map and empirical evidence in recent works (Farina et al., 2024; Ge et al., 2024), our method attempts to make agents effectively inherit and reuse each other's "thought processes," transcending the limitations of prefix-matched caching and enabling collaborative working memory sharing across agents.

#### 3.1.2 VALIDITY OF SEGMENT-LEVEL KV CACHE SHARING

To further validate the feasibility of segment-level KV cache sharing, we empirically demonstrate that the attention patterns of a given semantic segment remain highly consistent whether or not segment-level KV cache sharing is employed. When a new agent query contains a substring that matches a cached segment, even under entirely different prefix instructions, the corresponding KV cache can be seamlessly integrated into the new computational sequence. This effectively bypasses recomputation, as the cached attention states retain both predictive accuracy and functional integrity.

Fig. 1 compares attention maps during decoding with and without segment-level KV cache sharing. Specifically, taking the example in Sec. , Fig. 1 illustrates the attention map between $O_B$ and $Q_B = [Q_{B1}, O_A, Q_{B2}]$. The segment between the red lines corresponds to the attention map between $O_B$ and $O_A$. Three key observations emerge: **1)** The attention patterns in Fig. 1a (without segment-level KV sharing) and Fig. 1b (with segment-level KV sharing) exhibit strong similarity, motivating the direct reuse of originally computed KV cache for repeated semantic segments across

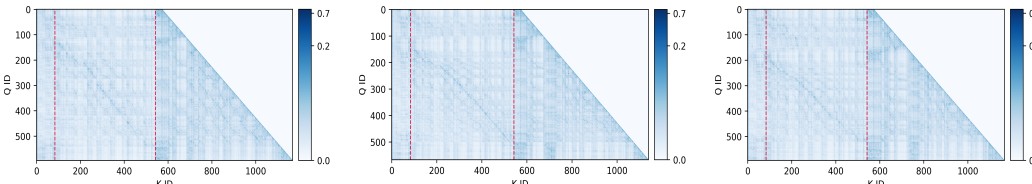

(a) Attention map w/o segment-level KV sharing

(b) Attention map w/ segment-level KV sharing

(c) Attention map w/ segment-level KV sharing and RoPE correction

Figure 1: Attention maps between output and the preceding tokens with and without KV cache sharing. We conduct experiments on Qwen2.5-7B-Instruct model. The segment between red lines indicates the tokens that can be reused. Detailed contexts are in Table 3 in the appendix.

different queries; **2)** The most significant attention values, i.e., those with larger magnitudes, are concentrated to some specific positoins, suggesting that partial recomputation may be beneficial in scenarios requiring high precision; **3)** Although reused segments may appear at different positional offsets, especially under rotary positional encoding (RoPE) (Su et al., 2024), the attention map after RoPE adjustment (Fig. 1c) remains similar to the unadjusted version, indicating that the effect of positional shift is negligible. We will further discuss on the impact of positional encoding in Sec. 3.3. Additionally, more experimental results on various agentic workflows confirm that cross-query KV cache reuse has minimal impact on instruction-following behavior, as shown and analyzed in Sec. 4.

### 3.2 EFFICIENT PROTOTYPE FOR CROSS-AGENT WORKING MEMORY SHARING

This section describes the implementation of our efficient cross-agent working memory sharing mechanism. Built upon the vLLM inference engine (Kwon et al., 2023) and its PagedAttention strategy, our prototype introduces a segment-level caching architecture that supports fine-grained KV cache reuse across diverse agent contexts. In the vLLM framework, the KV cache is organized into fixed-size blocks of consecutive tokens. Each block is assigned a logical hash value that serves as a unique identifier, computed based on the content of the current block and its preceding context. vLLM maintains a `Prefix Hash Table` to manage the mapping between these logical hash values and the physical addresses of the corresponding KV cache blocks.

To enable segment-level KV cache sharing, we introduce a `Memory Table` that stores mappings between the hash of a reusable semantic segment and the list of logical block hashes comprising that segment. The physical addresses of the corresponding KV cache blocks can then be retrieved by querying vLLM's built-in `Prefix Hash Table`. The inference process can be divided into three stages: (1) Pre-Alias during Pre-Filling: Reusable segments are identified and their KV cache blocks are logically aliased to new hashes without moving data; (2) Decoding: vLLM's PagedAttention strategy is employed for efficient autoregressive decoding; (3) Post-Recording: The KV caches of agent outputs are recorded into the `Memory Table` for future reuse. This design enables efficient retrieval and integration of cached semantic segments during both prefill and decoding stages, irrespective of variations in prefix instructions or overall sequence structure. By extending vLLM's native memory management to support reuse beyond exact prefix matches, our system significantly reduces computational redundancy while remaining fully compatible with existing attention algorithms. In the following, we will introduce the Pre-Alias and Post-Recording stages in detail.

#### 3.2.1 PRE-ALIAS DURING PRE-FILLING

During the pre-filling phase, the system identifies reusable KV cache segments and establishes logical aliases of the corresponding physical blocks to minimize redundant computation. Specifically, when a segment of text is enclosed between the special tags `<reuse_begin>` and `<reuse_end>`, it is flagged as a candidate for reuse. The system computes a hash value for this segment and queries the `Memory Table` to retrieve the historical list of logical block hashes associated with that segment, denoted as $L_{hist}$. Concurrently, the logical block hashes for the same segment in the current request are computed based on the latest context and prefix, forming a new list $L_{curr}$. Let PHT represent vLLM's internal `Prefix Hash Table`, which maps logical block hashes to physical block addresses. To enable reuse, we update this hash table such that:

$$\text{PHT}[L_{curr}[i]] = \text{PHT}[L_{hist}[i]], \quad \forall i \in [0, \text{len}(L_{hist}) - 1] \qquad (1)$$

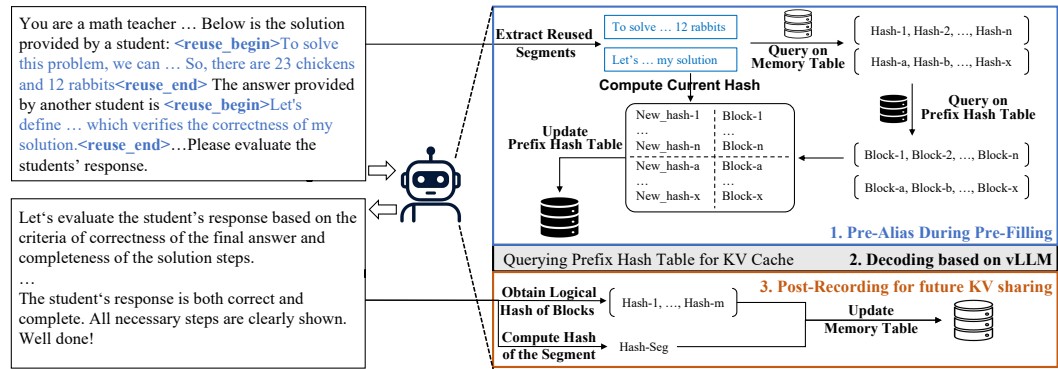

Figure 2: Overview of our CrossKV. Upon receiving a request, the agent first identifies reusable segments marked between `<reuse_begin>` and `<reuse_end>` tags in blue. (a) In the pre-filling stage, our method resolves the physical addresses of the corresponding KV cache blocks for these segments and performs Pre-Alias to map new logical hashes to these physical blocks in vLLM's built-in `Prefix Hash Table`. (b) In the decoding stage, vLLM's efficient attention mechanism is used to generate the response. (c) After the decoding stage, post-recording is applied and treats the agent's full output as a new segment: Hash of the the entire segment and the logical hashes of its constituent blocks are stored in the `Memory Table` to support future reuse by other agents.

This operation aliases the new logical hashes to the existing physical KV cache blocks of the historically computed segment, allowing subsequent attention computations to access the cached representations without recomputation.

### 3.2.2 POST-RECORDING

After the decoding phase completes, a post-recording step is performed to identify and store reusable segments from the agent's output for future cross-agent sharing. Our method supports two strategies for defining such reusable segments: The first strategy treats the agent's entire output as a single reusable segment. This approach is deterministic, requiring no additional model calls, and is directly governed by the predefined multi-agent workflow, making it robust and efficient. The second, more refined strategy involves prompting the LLM to automatically identify and mark semantically meaningful segments within its output using special tags (`<reuse_begin>` and `<reuse_end>`). This allows the agent to capture and reuse finer-grained semantic units, such as facts, reasoning steps, or plan fragments, that are more likely to be repeated across different contexts and agent interactions. However, this LLM-based strategy depends on the ability of LLM, which is not used in the present work and is discussed here as a potential direction for future exploration. We utilize the first strategy by default due to its efficiency and effectiveness. In the current implementation, we exclusively employ the first and primary strategy.

Under both strategies, a unique hash value representing the marked or full output segment is computed. This segment-level hash serves as the key in the `Memory Table`. The value associated with this key is the list of logical block hashes corresponding to the KV cache blocks of that segment, and each block contains a fixed number of tokens as defined in vLLM's memory layout. These logical block hashes are also stored in vLLM's internal `Prefix Hash Table`, which maintains mappings between logical hashes and physical KV cache addresses. Notably, the `Memory Table` itself only stores hash values rather than the actual KV cache, resulting in minimal memory overhead. The physical KV cache blocks are shared and reused across agents without duplication, ensuring that our mechanism introduces negligible additional storage and time cost.

By recording both the segment hash and its constituent block hashes, the `Memory Table` effectively links reusable semantic units to their cached computational states. When a future agent's query contains an identical segment, even within a novel contextual setting, our method can efficiently retrieve the corresponding KV cache blocks by querying the `Memory Table` and `Prefix Hash Table`. This process significantly reduces redundant computation while supporting dynamic, segment-level memory reuse across agents.

### 3.3 DISCUSSION ABOUT THE IMPACT OF POSITIONAL ENCODING

A critical technical issue in reusing KV cache segments involves the handling of positional encoding. Since the absolute positions of a reused segment in a new sequence generally differ from its original context, directly reusing its KV cache where keys are already applied with positional encoding may introduce positional misalignment and potentially affect contextual coherence. Since RoPE is widely used in most of the LLMs, we analyze its impact in detail. In principle, the positional shift could be corrected by recomputing the RoPE for the reused keys based on their new positions. Specifically, the attention of $q$ at position $m$ and $k$ at position $n$ with RoPE can be computed as follows:

$$\text{Attn}(q,k) = (R_m q)^\top (R_n k) = q^\top R_{n-m} k, \tag{2}$$

where $R_m$ indicates the rotation matrix at position $m$, and $R_m^\top R_n = R_{-m} R_n = R_{n-m}$ according to the property of rotation matrix. Suppose $\hat{k}_n = R_n k$ denotes the reused key vector that has been aligned by RoPE at position $n$ at early stage. In the current decoding stage, the reused key vector is located at position $n'$, then the vector after RoPE alignment can be computed as:

$$\hat{k}_{n'} = R_{n'} k = R_{n'-n} R_n k = R_{n'-n} \hat{k}_n. \tag{3}$$

Hence, we can correct the positional shift by conducting RoPE alignment once. However, it is important to note that applying RoPE correction requires creating copies of the KV cache blocks and performing the rotational transformation, resulting in additional memory I/O and computational overhead. Table 1 illustrates the inference effi-

Table 1: TTFT (Time-To-First-Token) Comparison Across Methods.

| #Reused tokens | Vanilla | CrossKV | CrossKV+RoPE |
|---|---|---|---|
| 0.5k | 130.34 ms | 92.73 ms | 105.33 ms |
| 2k | 278.28 ms | 113.25 ms | 128.07 ms |
| 8k | 818.89 ms | 114.76 ms | 131.86 ms |

ciency, measured by Time-To-First-Token (TTFT), which reflects the latency of the prefill stage. In fact, as illustrated in Fig. 1b and Fig. 1c, visual analysis of attention patterns suggests that the effect of positional shift is often negligible. This observation is further corroborated by our experiments on agentic workflows in Sec. 4, where directly sharing segment-level KV cache exhibits negligible impact on performance in most common scenarios.

In order to assess the cases where positional correction counts, we further conducted targeted experiments and found that when instructions are complex and very long segments are reused, uncorrected positional misalignment can indeed impair instruction-following capability. Detailed examples are shown in the appendix A.3. Interestingly, we find that partial recomputation of the segment, where only a portion of tokens is recomputed rather than the entire segment, can also help mitigate this issue. Since the recomputed tokens receive correct positional encoding during their generation, they effectively "anchor" the segment with accurate positional information. This suggests that recomputation serves as an alternative strategy to address positional misalignment while avoiding the full cost of RoPE correction.

Based on this insight, we propose an adaptive recomputation strategy that dynamically determines when to apply partial recomputation based on segment length. Specifically, when the reused segment exceeds a certain token threshold, the system automatically triggers recomputation of a percentage of tokens at the beginning. This approach balances the benefits of positional accuracy with computational efficiency, making it suitable for practical deployment. Nevertheless, since both approaches introduce additional overhead, and most practical settings do not exhibit significant degradation, we disable positional correction and set the maximum sharing length for a segment as $N$ tokens ($N = 1600$ in our experiments to balance the speed and accuracy). When a segment exceeds this length threshold, our system automatically applies an adaptive reuse strategy: it reuses the last $N$ tokens of the segment while recomputing the initial portion. This approach effectively balances inference speed and output accuracy for applications requiring high precision with long contextual segments.

## 4 EXPERIMENTS

Our method is evaluated across a diverse range of models, benchmarks, and agentic workflows. Specifically, the models span multiple scales and architectural families, including: Qwen2.5-7B-Instruct, Qwen2.5-72B-Instruct (Bai et al., 2023), and Llama-3.1-8B-Instruct (Dubey et al., 2024);

Table 2: Effectiveness of segment-level KV cache sharing for agentic workflows. The table compares task accuracy between the baseline (vanilla workflow) and our proposed CrossKV across various models and benchmarks. We also report the results of the proposed re-computation strategy, denoted as 'CrossKV-R'. Cells in gray indicate the best performance under the same setting.

| Model | Tasks | AutoGen | | | MAD | | | Solver | | |
|---|---|---|---|---|---|---|---|---|---|---|
| | | Vanilla | CrossKV | CrossKV-R | Vanilla | CrossKV | CrossKV-R | Vanilla | CrossKV | CrossKV-R |
| Qwen2.5-7B Instruct | GSM-Hard | **54.20** | 50.60 | 53.00 | **58.60** | 56.00 | 54.20 | **55.40** | 55.20 | 44.60 |
| | MATH | **75.60** | 74.60 | 73.00 | **76.16** | 75.80 | 75.40 | **72.80** | 68.20 | 66.40 |
| | GPQA | **30.36** | 28.12 | 29.02 | 25.00 | **30.13** | 25.67 | 24.78 | **29.91** | 24.11 |
| | AQUA-RAT | 73.62 | 70.08 | **74.41** | 79.13 | **80.31** | 75.59 | 69.29 | **69.29** | 62.60 |
| | MMLU-Pro | **55.20** | 53.00 | 52.80 | 42.20 | 46.20 | **50.00** | 48.60 | 45.60 | 42.80 |
| | SciBench | 20.24 | **21.04** | 18.44 | **21.04** | 20.84 | 20.24 | **20.44** | 16.23 | 16.43 |
| | MedQA | 57.40 | **57.60** | **57.60** | 33.20 | 46.40 | **53.20** | 50.80 | **54.00** | 53.20 |
| | MedMCQA | **55.00** | 51.60 | 53.60 | 45.60 | 52.00 | **52.40** | 52.60 | **55.20** | 52.20 |
| Qwen2.5-72B Instruct | GSM-Hard | 65.00 | **65.80** | 64.40 | 62.40 | **63.60** | 62.40 | 62.20 | **64.40** | 59.60 |
| | MATH | **82.00** | 79.80 | 79.20 | 83.00 | **83.80** | 82.40 | **82.80** | 78.80 | 80.00 |
| | GPQA | 43.08 | **45.54** | 42.86 | **47.32** | 42.41 | 44.42 | **43.97** | 43.53 | 43.53 |
| | AQUA-RAT | 77.56 | **80.31** | 78.35 | 79.53 | 79.53 | **79.92** | **79.92** | 78.35 | 75.59 |
| | MMLU-Pro | 69.40 | 68.40 | **71.20** | 65.80 | 67.60 | **68.40** | **69.60** | 66.40 | 61.00 |
| | SciBench | 25.05 | **26.05** | **26.05** | 27.25 | 26.85 | **27.45** | **26.45** | 25.45 | 24.45 |
| | MedQA | 78.40 | 77.80 | **79.80** | **78.40** | 75.80 | 78.20 | **80.00** | 79.80 | 76.40 |
| | MedMCQA | **70.20** | 68.00 | 68.40 | 66.40 | **67.60** | 64.20 | 69.00 | **70.00** | 67.40 |
| Llama3-8B Instruct | GSM-Hard | 14.60 | 12.00 | **25.60** | 22.60 | 29.40 | **28.00** | 20.00 | **28.00** | 26.00 |
| | MATH | 16.80 | 15.60 | **35.80** | 33.80 | **38.40** | 37.20 | 29.00 | **40.80** | 37.20 |
| | GPQA | 13.62 | 10.49 | **23.66** | 24.11 | 23.44 | **24.78** | 20.98 | 17.63 | **23.21** |
| | AQUA-RAT | 48.82 | **51.18** | 50.00 | 46.85 | 58.66 | **59.45** | 40.94 | 42.53 | **48.82** |
| | MMLU-Pro | 28.60 | **33.40** | 30.20 | 27.80 | 31.40 | **31.80** | **40.20** | 36.60 | 37.40 |
| | SciBench | 6.01 | 4.21 | **8.62** | 11.02 | **11.22** | **11.22** | **9.42** | 8.62 | 7.82 |
| | MedQA | 43.60 | 52.40 | **56.40** | 23.20 | 40.40 | **41.80** | 53.60 | 49.40 | **53.60** |
| | MedMCQA | 41.60 | **47.20** | 45.80 | 25.80 | **36.40** | 34.80 | **50.20** | 47.40 | **50.20** |

Our experiments encompass a broad spectrum of domains to ensure comprehensive validation, including MATH (Hendrycks et al., 2021), GSM-Hard (Gao et al., 2023), and AQUA-RAT (Ling et al., 2017) for mathematical reasoning using; SciBench (Wang et al., 2023) and GPQA (Rein et al., 2024) for scientific reasoning; MMLU-Pro (Wang et al., 2024) for general knowledge comprehension; and MedQA (Jin et al., 2021) and MedMCQA (Pal et al., 2022) for medical domain performance. All evaluations are conducted within the MASLab framework (Ye et al., 2025), a unified benchmarking platform for multi-agent systems. To demonstrate practical utility, we integrate our approach into established multi-agent frameworks, including AutoGen (Wu et al., 2024) and MAD (Liang et al., 2023), showcasing its applicability and effectiveness in realistic collaborative environments. Additionally, we have summarized the workflows of previous agent system (Wooldridge & Jennings, 1995; Hong et al., 2024; Zhao et al., 2025; Parmar et al., 2025; Chai et al., 2025) and derived the Analysis-Solution-Verification paradigm, denoted as Solver in our experiments. Detailed introduction to this workflow can be found in the appendix A.1.

## 4.1 RESULTS ON AGENTIC WORKFLOWS

Table 2 presents the comprehensive experimental results, showing notable effectiveness of our CrossKV across multiple models, tasks, and multi-agent workflows. Overall, the method leads to performance improvements in a majority of task–framework combinations, particularly under the MAD workflows. For instance, using Qwen2.5-7B Instruct on MedQA under the MAD workflow, CrossKV achieves a significant improvement of +13.20%. Similarly, with Llama3-8B Instruct, substantial gains are observed across several tasks: +17.20% on MedQA (MAD), +11.81% on AQUA-RAT (MAD), and +11.80% on MATH (Solver). These results suggest that efficient reuse of computation can lead to noticeable performance uplifts.

While there are occasional performance regressions, such as with Qwen2.5-7B on GSM-Hard across all frameworks—these are often marginal and are outweighed by the positive results in more complex and knowledge, intensive tasks like GPQA, AQUA-RAT, and MedMCQA. Moreover, the larger model (Qwen2.5-72B Instruct) maintains highly competitive performance with minor fluctuations, indicating that the method preserves output quality while improving efficiency. The consistent improvements across heterogeneous tasks and agent frameworks underscore the general applicability and robustness of the proposed KV cache reuse strategy, affirming its potential as a system-level enhancement for collaborative multi-agent reasoning.

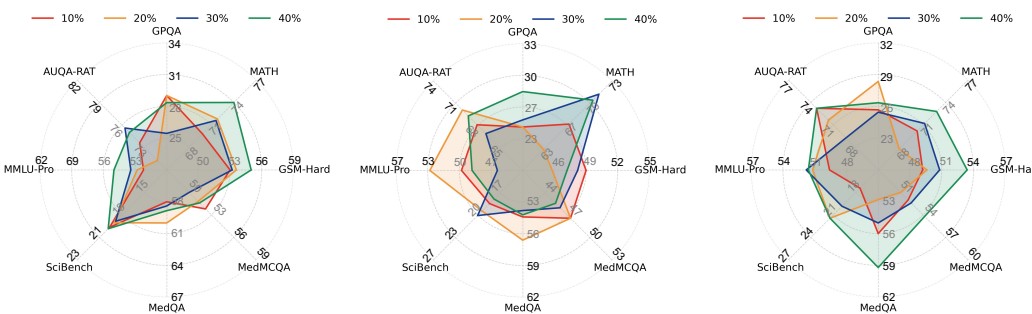

(a) Recompute the KV cache for the first several tokens.

(b) Recompute the KV cache for both the first and last several tokens

(c) Recompute the KV cache for the last several tokens.

Figure 4: Evaluation of three KV cache recomputation strategies on various benchmarks. Experiments are conduct on the Qwen2.5-7B-Instruct model and AutoGen workflow.

## 4.2 ANALYSIS ON INFERENCE SPEED

Fig. 3 compares the cache hit rate and inference speed between CrossKV and the vanilla approach, demonstrating that our segment-level KV cache sharing strategy yields significant improvements. The proposed method consistently achieves higher hit rates across varying sequence lengths, confirming its effectiveness in identifying and leveraging reusable semantic segments. In addition, prefill speed is substantially enhanced, highlighting the mechanism's ability to minimize redundant computation and accelerate initial processing. These efficiency gains validate

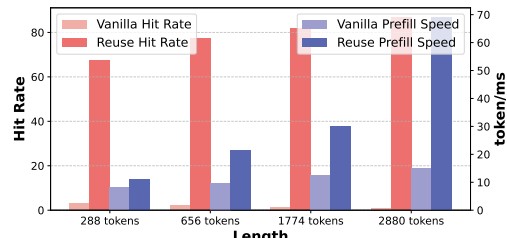

Figure 3: Cache hit rate and prefill speed (inverse of TTFT) with and without segment-level KV cache sharing for one request.

the practical utility of our segment-level caching approach in boosting multi-agent inference while ensuring high cache utilization. Detailed time cost measurements across multiple benchmarks are provided in Appendix A.5, offering further evidence of the method's effectiveness.

## 4.3 INFLUENCE OF RECOMPUTE RATE

Recent studies (Hu et al., 2024; Yao et al., 2025) suggest that partial recomputation of the KV cache can enhance the accuracy. Hence we adopted three straightforward recomputation strategies inspired by EPIC Hu et al. (2024): recomputing the KV cache only for the first several tokens, for the last several tokens, and for both the first and last several tokens. As illustrated in Fig. 4, we observe that recomputing the first several tokens generally yields better performance, and increasing the number of recomputed tokens correlates with moderate improvements in multi-agent reasoning accuracy. However, this gain comes at the cost of reduced inference speed, highlighting a non-trivial trade-off. It is worth noting that even without recomputation, our method already demonstrates strong performance advantages, as evidenced in Table 2. Detailed accuracy values corresponding to these recomputation strategies are provided in the appendix A.4.

## 5 CONCLUSION

This paper addresses the under-explored potential of leveraging KV cache as a working memory carrier in agentic workflow. Moving beyond the rigid prefix-matching mechanism, we introduce a novel segment-level KV cache sharing mechanism that allows dynamic and segment-level reuse of semantic segments across agents. This approach not only accelerates inference but also enables cross-agent working memory sharing. A high-performance prototype is further developed based on the vLLM inference engine, demonstrating the practical feasibility of our method under realistic multi-agent workloads. Extensive experiments on a range of models and benchmarks show that our method significantly improves inference efficiency while also enhancing task performance across several multi-agent workflows. These gains confirm that our framework is not only an optimization

tool but also a means of strengthening collaborative intelligence. Additionally, we analyzed important technical considerations such as the effect of positional encoding (RoPE) and the trade-offs involved in partial KV cache recomputation. We believe this study provides an important foundation for building efficient and scalable multi-agent collaboration and promotes the development of next-generation KV cache sharing strategies for heterogeneous collaborative scenarios.

**Limitations and Future Work.** Although this work demonstrates the effectiveness of segment-level KV cache sharing in multi-agent collaboration, it currently lacks a theoretical analysis of why segment-level cache reuse is successful. Although providing such formal guarantees is challenging due to the inherent interpretability limitations of LLM, we acknowledge this as a valuable direction and would like to leave it as our future work. Additionally, more efficient strategies for handling positional encoding and optimized recomputation policies also represent promising avenues for further exploration, which could enhance performance in long-context and complex multi-agent scenarios.

## ETHICS STATEMENT

This work focuses on improving the efficiency and performance of KV cache sharing in multi-agent systems. All experiments were conducted on publicly available datasets and within simulated environments, involving no private or sensitive data. The benchmark datasets used (e.g., MATH, GSM-Hard, MedQA) are standard academic benchmarks and raise no ethical concerns. Our research did not involve any human subjects or applications that could lead to issues of bias, discrimination, or social harm. We note that efficient KV cache sharing has the potential to reduce the computational cost and energy consumption of large language model inference, offering potential environmental benefits. We are committed to the responsible development and deployment of this technology and will continue to consider its ethical implications in practical scenarios.

## REPRODUCIBILITY STATEMENT

We will open-source our implementation code to promote reproducibility. Our proposed system, CrossKV, is implemented on top of the open-source vLLM inference engine. The segment-level KV cache sharing mechanism is comprehensively described in Sec. 3, with detailed explanations of its three core stages: Pre-Alias during pre-filling, Decoding, and Post-Recording. All experimental configurations are clearly outlined, including the models employed (e.g., Qwen-2.5, Llama-3.1), benchmark tasks (e.g., MATH, GPQA, MMLU-Pro), multi-agent frameworks (e.g., AutoGen, MAD, Solver), and evaluation metrics. Key hyperparameters, cache reuse strategies, such as maximum segment length and partial recomputation ratios, and the handling of positional encodings (e.g., RoPE correction) are thoroughly documented. Additionally, the appendix provides detailed query-output examples, attention map visualizations, and time-cost analyses to support the replication and validation of our findings.

## DECLARATION OF AI USE

We declare that LLM was used during the writing of this paper to assist in translating and polishing the English text. However, all intellectual content, including the core ideas, analyses, and final conclusions, remains entirely our own.

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

# A APPENDIX

## A.1 DETAILS OF AGENT WORKFLOWS IN OUR EXPERIMENTS

**AutoGen** AutoGen processes user queries through interactions between the User Proxy Agent and the Assistant Agent. The workflow starts with receiving a user query, which the Assistant Agent uses to generate an initial response. It then enters a multi-turn conversation loop: in each turn, the User Proxy Agent first receives the response from the Assistant Agent. If code execution is enabled, it runs the code contained in the response and returns the execution result. Afterward, the User

Proxy Agent generates a new request and feeds it back to the Assistant Agent. This loop continues until the maximum number of turns is reached or a termination condition is triggered.

**MAD** This workflow implements a Multi-Agent Debate (MAD) system, which primarily completes debate tasks through interactions among three types of agents: "Affirmative side", "Negative side", and "Moderator". If the debate fails to meet termination conditions, a "Judge" is introduced to generate the final result. The process starts with receiving a debate topic, first initializing the prompts for each agent, such as replacing topic placeholders and creating agent instances. It then enters the debate initiation phase, where the affirmative side puts forward arguments based on preset prompts, the negative side responds to the affirmative's arguments, and the moderator evaluates the first-round content from both sides and generates feedback in JSON format. Next, it proceeds to a multi-round debate loop. In each round, the affirmative and negative sides sequentially supplement their arguments in response to the opponent's previous round of views, while the moderator updates the evaluation according to the current round, until the maximum number of rounds is reached or the moderator determines the debate can be terminated. If the moderator still hasn't given a final conclusion after the loop ends, a judge agent is created. The judge evaluates based on all debate content from both sides and generates the final debate result, which is then returned as the output.

**Solver** Drawing on multi-agent workflow designs, we have developed a "Solver" framework. Specifically, this workflow comprises three distinct agents: the Analyst, Solver, and Verifier, each with dedicated responsibilities. First, the Analyst conducts a comprehensive analysis of the target problem and formulates a general solution framework. Next, the Solver takes over this framework and executes step-by-step problem-solving based on it. Finally, the Verifier reviews the Solver's output, identifies any errors, and makes necessary corrections.

## A.2 INSTANCE OF QUERY AND OUTPUTS

Table 3 and Table 4 present two examples in which reusing the KV cache of the entire segment has only a minimal impact on generation performance, even if the positional encoding of the reused KV cache is shifted.

## A.3 INSTANCE OF QUERY FOR RoPE CORRECTION

In order to assess the cases where positional correction is impactful, we conducted targeted experiments and observed that instruction-following capability can indeed be impaired when complex instructions are coupled with the reuse of very long segments under uncorrected positional misalignment.

Specifically, we designed a cross-lingual instruction scenario: the first agent received an English instruction asking it to generate a survey on KV cache (no less than 5000 words), while the second agent was instructed in Chinese to review the generated survey and produce a Chinese evaluation. As shown in Table 5, reusing the segment-level KV cache in this case led the second agent to ignore its original instruction and instead continue writing the survey—clearly deviating from the intended task. While, recomputing 40% tokens will address the above issue.

To verify whether the issue stems from the complexity of mixed-language instructions combined with segment length, we carried out three controlled experiments: 1) Mixed Chinese and English instructions, but with the survey limited to 1000 words (Table 6). 2) Both instructions in Chinese, with similar length requirement (Table 7); 3) Both instructions in English, with the survey requirement exceeding 5000 words (Table 8);

Results confirm that under these settings, segment-level KV cache reuse maintains strong instruction compliance and produces outputs consistent with task requirements. This suggests that the degradation observed earlier arises specifically under conditions involving both high instructional complexity (e.g., cross-lingual cues) and long-context reuse.

Table 3: Example 1. The instructions are in red, and the reusable segment between `reuse_begin` and `reuse_end` tags is generated before current request.

| Input | Vanilla | CrossKV |
|---|---|---|
| You are a math teacher, and here is a problem for your class: Put chickens and rabbits in the same cage. There are 35 heads and 94 legs in all. How many chickens and how many rabbits are there? Below is the solution provided by a student:
**\<reuse_begin\>**To solve this problem, we can use a system of linear equations based on the information given:
1. Let's define: - $C$ as the number of chickens. - $R$ as the number of rabbits.
2. We know that each chicken has 1 head and 2 legs, and each rabbit has 1 head and 4 legs. Therefore, we can set up the following equations based on the total number of heads and legs:
**Equation for the number of heads:**
$$C + R = 35$$
**Equation for the number of legs:**
$$2C + 4R = 94$$
3. To simplify the second equation, we can divide it by 2:
$$C + 2R = 47$$
4. Now we have the simplified system of equations:
$$\begin{cases} C + R = 35 \\ C + 2R = 47 \end{cases}$$
5. Subtract the first equation from the second equation to eliminate $C$:
$$(C + 2R) - (C + R) = 47 - 35$$
Simplifying this, we get:
$$R = 12$$
6. Substitute $R = 12$ back into the first equation to find $C$:
$$C + 12 = 35$$
Solving for $C$:
$$C = 35 - 12 = 23$$
So, there are 23 chickens and 12 rabbits.
**Verification:** - Total number of heads: $23 + 12 = 35$ - Total number of legs: $2 \times 23 + 4 \times 12 = 46 + 48 = 94$
Both conditions are satisfied, so our solution is correct.**\<reuse_end\>**
Please evaluate the student's response based on the following criteria: correctness of the final answer and completeness of the solution steps. | The student's response is correct and well-structured. Here is an evaluation based on the given criteria:
**Correctness of the Final Answer:**
- The student correctly identified that there are 23 chickens and 12 rabbits.
- Verification was done to check if the solution satisfies both the number of heads and legs:
- Total number of heads: $23 + 12 = 35$
- Total number of legs: $2 \times 23 + 4 \times 12 = 46 + 48 = 94$
- Both conditions are satisfied, confirming the solution is correct.
**Completeness of the Solution Steps:**
- Step 1: Defined variables $C$ (number of chickens) and $R$ (number of rabbits).
- Step 2: Set up the system of linear equations based on the given information:
- Equation for the number of heads: $C + R = 35$
- Equation for the number of legs: $2C + 4R = 94$
- Step 3: Simplified the second equation by dividing by 2 to get $C + 2R = 47$.
- Step 4: Set up the simplified system of equations:
$$\begin{cases} C + R = 35 \\ C + 2R = 47 \end{cases}$$
- Step 5: Eliminated $C$ by subtracting the first equation from the second:
$$(C+2R)-(C+R) = 47-35 \implies R = 12$$
- Step 6: Substituted $R = 12$ back into the first equation to find $C$:
$$C + 12 = 35 \implies C = 23$$
**Additional Feedback:**
- The solution is clear and logical.
- The steps are well-explained and easy to follow.
- The verification step is included, which adds credibility to the solution.
Overall, the student's response is thorough, accurate, and well-structured. There are no significant issues or omissions in the solution process. | Certainly! Let's evaluate the student's response based on the criteria of correctness of the final answer and completeness of the solution steps.
**Correctness of the Final Answer** The student found that there are 23 chickens and 12 rabbits. Let's verify this:
- Number of Heads:
$$23 \text{ chickens} + 12 \text{ rabbits} = 35 \text{ heads}$$
- Number of Legs:
$$2 \times 23 \text{ (chickens)} + 4 \times 12 \text{ (rabbits)}$$
$$= 46 + 48 = 94 \text{ legs}$$
Both the number of heads and the number of legs match the given conditions, so the final answer is correct.
**Completeness of the Solution Steps**
Now let's check if the solution steps are complete and clear:
1. Define Variables:
- The student correctly defined $C$ as the number of chickens and $R$ as the number of rabbits.
2. Set Up Equations:
- The student correctly set up two equations based on the given information:
$$C + R = 35 \quad \text{(number of heads)}$$
$$2C + 4R = 94 \quad \text{(number of legs)}$$
3. Simplify the Second Equation:
- The student divided the second equation by 2 to simplify it:
$$C + 2R = 47$$
4. Eliminate One Variable:
- The student subtracted the first equation from the simplified second equation to eliminate $C$:
$$(C + 2R) - (C + R) = 47 - 35$$
$$R = 12$$
5. Substitute Back to Find the Other Variable:
- The student correctly substituted $R = 12$ back into the first equation to find $C$:
$$C + 12 = 35$$
$$C = 23$$
6. Verification:
- The student verified the solution by checking both the number of heads and legs, which confirms the correctness of the answer.
**Conclusion**
The student's response is both correct and complete. All necessary steps are clearly shown, and the final answer matches the given conditions. Well done! |

Table 4: Example 2. The instructions are in red, and the reusable segment between `reuse_begin` and `reuse_end` tags is generated before current request.

| Input | Vanilla | CrossKV |
|---|---|---|
| You are a travel expert. Below is a travel plan that you will be evaluating. Please evaluate the plan based on its content without making any modifications to the text. Provide your evaluation directly. Assess the plan from the perspectives of completeness and feasibility, and assign a score on a percentage basis. The plan you need to evaluate is as follows: **<reuse_begin>**Certainly! Here's a detailed travel plan for a family trip during Christmas, tailored to provide a memorable and enjoyable experience: Family Christmas Travel Plan Destination: Orlando, Florida - **Reasons for Choosing Orlando**: Known for its numerous theme parks, festive decorations, and family-friendly attractions. — Week 1: Planning and Preparations **Day 1:** - **Morning**: Family meeting to finalize the itinerary and discuss expectations. - **Afternoon**: Book flights and accommodations (e.g., Disney's Grand Floridian Resort & Spa). - **Evening**: Pack essentials and review travel documents. **Day 2:** - **Morning**: Depart from home. - **Afternoon**: Arrive in Orlando; check into hotel. - **Evening**: Relax and explore the area nearby. — Week 2: Activities and Attractions **Day 3:** - **Morning**: Breakfast at the hotel. - **Afternoon**: Visit Universal Studios or Islands of Adventure. - **Evening**: Return to hotel; enjoy dinner at a themed restaurant. **Day 4:** - **Morning**: Breakfast at the hotel. - **Afternoon**: Explore SeaWorld Orlando. - **Evening**: Return to hotel; relax and enjoy the evening. **Day 5:** - **Morning**: Breakfast at the hotel. - **Afternoon**: Visit the Magic Kingdom. - **Evening**: Attend a fireworks show at Magic Kingdom. **Day 6:** - **Morning**: Breakfast at the hotel. - **Afternoon**: Visit Epcot or Disney Springs. - **Evening**: Return to hotel; enjoy a special Christmas dinner. **Day 7:** - **Morning**: Breakfast at the hotel. - **Afternoon**: Free time to explore or visit other nearby attractions. - **Evening**: Return home. — Additional Tips - **Transportation**: Consider renting a car for flexibility, especially if you have younger children who might need more space. - **Accommodation**: Choose a resort with amenities like pools, spas, and dining options to make your stay more comfortable. - **Theme Park Tickets**: Purchase tickets in advance to save money and avoid long lines. - **Safety Measures**: Check the latest health guidelines and safety protocols before traveling. - **Weather**: Orlando can be warm even in December, so pack accordingly. Layers are recommended as it can cool down in the evenings. Conclusion This itinerary provides a mix of fun, relaxation, and festive activities that will make your Christmas trip memorable for the whole family. Enjoy your holiday adventure in Orlando!**<reuse_end>** Do not continue writing any content; output your evaluation directly. | Evaluation Completeness: The travel plan is well-structured and covers all essential aspects of a family Christmas trip to Orlando. It includes detailed daily schedules, specific activities, and additional tips. The plan is comprehensive and addresses various needs such as transportation, accommodation, theme park tickets, and safety measures. The inclusion of weather advice and packing recommendations ensures that families are prepared for the trip. Feasibility: The plan is highly feasible and practical. It accounts for the family's needs and preferences, including the choice of a family-friendly destination and a variety of activities suitable for different ages. The itinerary is realistic and allows for adequate rest days and free time. The suggestions for purchasing tickets in advance and renting a car are practical and can help manage logistics effectively. Score: 95% The plan is thorough and well-organized, addressing all necessary aspects of a family Christmas trip to Orlando. With only minor adjustments, this plan would be an excellent guide for a memorable and enjoyable family vacation. | Evaluation Completeness: The travel plan is comprehensive and covers various aspects of the trip, including transportation, accommodation, daily activities, and meal recommendations. However, it lacks some details such as specific times for activities, exact dates, and potential backup plans in case of weather-related issues or park closures. Additionally, there is no mention of entertainment options within the hotel or local areas outside of the theme parks. Feasibility: The plan is feasible given the chosen destination and activities. Orlando is known for its family-friendly attractions, and the proposed itinerary allows for a balanced mix of theme park visits and downtime. However, the plan may be too packed, potentially leading to fatigue, especially for younger children. It would be advisable to include more free time or lighter activities on one or two days. Score: 85% The plan is well-structured and covers most essential aspects of a family Christmas trip to Orlando. A few minor adjustments could enhance its completeness and feasibility. |

Table 5: Reusing all KV caches of a long segment will fail to follow the complex instruction (cross-lingual cues). The instructions are in red, and the reusable segment between `reuse_begin` and `reuse_end` tags is generated before current request.

| Input (5136 tokens can be reused) | 你是一个人工智能领域的专家，请你从内容的完整性、连贯性、准确性三个角度，用英文评审一下下面这段综述： |
| --- | --- |
| | **\<reuse_begin\>**Introduction to KV Cache in Large Language Models |
| | Key-Value (KV) Cache is a crucial technique in optimizing the performance of large language models (LLMs), particularly in the context of natural language processing tasks such as text generation, question answering, and dialogue systems. The primary function of KV Cache is to store intermediate results or states from previous model computations, allowing for faster and more efficient processing of subsequent inputs. This technique significantly enhances the throughput and latency of LLMs without compromising their accuracy, making it indispensable in real-time applications where quick responses are essential. |
| | In the realm of LLMs, the computational complexity involved in generating text or performing inference can be quite substantial. Each token generated requires the model to process a significant amount of information from its input sequence and its internal state. Traditional approaches to handling this complexity often involve extensive memory usage and slower inference times, which can become prohibitive for large models operating at high speeds. KV Cache addresses these challenges by caching key-value pairs that represent the model's internal state, thereby reducing redundant computations and accelerating the overall process. |
| | · · · |
| | Conclusion and Future Directions |
| | In conclusion, Key-Value (KV) Cache is a pivotal technique in the optimization of large language models (LLMs), offering substantial benefits in terms of performance, efficiency, and scalability. By caching key-value pairs representing intermediate results, KV Cache significantly reduces computational load, improves inference speed, and optimizes memory usage. This makes it an indispensable tool for real-time applications and large-scale deployments. |
| | However, the implementation of KV Cache also presents challenges such as increased memory usage, the need for effective cache management strategies, and the potential impact on model accuracy. To fully realize the benefits of KV Cache, it is crucial to follow best practices, including selecting appropriate data structures, optimizing cache eviction policies, ensuring efficient cache operations, and implementing distributed cache management strategies. Periodic validation and refreshing of cache contents are also essential to maintain accuracy. |
| | Looking ahead, future research and development in this area will likely focus on refining cache management techniques, exploring new data structures and algorithms, and addressing the integration of KV Cache in more complex and diverse applications. As LLMs continue to evolve, the role of KV Cache in enhancing their performance and efficiency will undoubtedly become even more critical. References |
| | - Google AI. (2019). **T5: A Text-to-Text Pretraining Model**. Retrieved from [Google AI Blog](https://ai.googleblog.com/2019/10/t5-google-s-text-to-text-next-generation.html) |
| | - Lee, J., Kim, Y., & Park, J. (2021). **Optimizing Transformer Models with Key-Value Cache**. *Proceedings of the International Conference on Learning Representations (ICLR)*. |
| | - Alibaba Cloud. (n.d.). **MaxCompute: Scalable Data Processing Platform**. Retrieved from [Alibaba Cloud Documentation](https://www.alibabacloud.com/product/maxcompute) |
| | This comprehensive review provides a detailed understanding of KV Cache in LLMs, highlighting its significance, challenges, and practical applications.**\<reuse_end\>** 仅需要写评审即可，不要输出其他内容。 |
| **Vanilla** | Content Completeness |
| | - **Introduction**: The introduction adequately covers the purpose and importance of KV Cache in LLMs. It provides a clear overview of the technique and its benefits. - **Technical Overview**: The technical section is well-detailed, explaining the mechanism behind KV Cache, including the use of key-value pairs, data structures, and cache management strategies. - **Benefits**: The benefits section comprehensively outlines the advantages of using KV Cache, such as reduced computational load, improved inference speed, and memory optimization. - **Challenges**: The challenges section effectively highlights potential issues like increased memory usage, cache hit rate, and latency, along with the importance of careful management. - **Best Practices**: The best practices section provides actionable advice on how to implement KV Cache effectively, covering data structures, cache eviction policies, and distributed cache management. - **Case Studies**: The case studies section includes relevant examples from industry and academia, demonstrating the practical application and effectiveness of KV Cache. - **Conclusion and Future Directions**: The conclusion summarizes the key points and discusses future research directions, providing a clear outlook for the field. |
| | Content Concurrency |
| | - **Introduction to Technical Overview**: The transition from the introduction to the technical overview is smooth and logical. The introduction sets the stage for the technical details that follow. - **Technical Overview to Benefits**: The shift from the technical explanation to the benefits is well-motivated. The technical details naturally lead to discussions about the benefits of using KV Cache. - **Benefits to Challenges**: The move from discussing benefits to challenges is logical and necessary. It provides a balanced view of the technique, highlighting both its strengths and limitations. - **Challenges to Best Practices**: The discussion on challenges leads naturally into best practices, as the latter section offers solutions and recommendations to mitigate the identified issues. - **Best Practices to Case Studies**: The best practices are followed by case studies, which provide concrete examples of how KV Cache is implemented and its practical outcomes. - **Case Studies to Conclusion**: The case studies serve as a bridge to the conclusion, reinforcing the practical benefits of KV Cache before summarizing the key takeaways. |
| | Content Accuracy |
| | - **Technical Details**: The technical details provided are accurate and well-explained. Concepts such as key-value pairs, data structures, and cache management strategies are clearly described. - **Benefits and Challenges**: The benefits and challenges discussed are accurate and relevant. The examples and explanations provided are consistent with the technical details. - **Best Practices**: The best practices suggested are accurate and well-founded. The recommendations align with the technical aspects and challenges discussed. - **Case Studies**: The case studies are accurately described and provide relevant examples. The sources cited are credible and provide additional context. - **References**: The references are correctly formatted and include relevant sources, ensuring the information is well-supported. |
| | Overall, the review is comprehensive, well-structured, and accurate in its presentation of KV Cache in LLMs. |

Table 5: Reusing all KV caches of a long segment will fail to follow the complex instruction (Continue). However, RoPE correction and recomputation strategy can address the issue.

| **CrossKV** (reuse all KV caches of the segment) | Conclusion and Future Directions
In conclusion, Key-Value (KV) Cache is a pivotal technique in the optimization of large language models (LLMs), offering substantial benefits in terms of performance, efficiency, and scalability. By caching key-value pairs representing intermediate results, KV Cache significantly reduces computational load, improves inference speed, and optimizes memory usage. This makes it an indispensable tool for real-time applications and large-scale deployments.
Key Benefits: 1. **Reduced Computational Load**: By caching intermediate results, the model avoids redundant calculations, leading to more efficient processing. 2. **Improved Inference Speed**: Rapid retrieval of precomputed hidden states allows for faster responses, enhancing the model's responsiveness. 3. **Enhanced Memory Optimization**: Efficient memory usage ensures that the model can handle larger input sequences and more complex tasks without performance degradation. 4. **Increased Robustness**: Caching maintains a record of previously computed states, aiding recovery from errors or interruptions and ensuring stable operation.
Challenges and Limitations: 1. **Increased Memory Usage**: Managing the cache size effectively is crucial to avoid excessive memory consumption. 2. **Cache Hit Rate and Eviction Policies**: Balancing cache hit rate and managing eviction policies to ensure optimal performance. 3. **Latency**: Minimizing the overhead of cache operations to reduce latency. 4. **Scalability**: Coordinating cache management across multiple nodes in distributed settings. 5. **Impact on Model Accuracy**: Ensuring that the cache does not introduce subtle inaccuracies and that the model's state is accurately reflected.
Best Practices for Implementation: 1. **Select Appropriate Data Structures**: Choose hash tables, dictionaries, or other efficient data structures based on specific requirements. 2. **Optimize Cache Eviction Policies**: Use LRU or LFU algorithms to manage cache size effectively. 3. **Ensure Efficient Cache Operations**: Optimize search and update operations to minimize latency. 4. **Implement Distributed Cache Management**: Use distributed cache solutions like Redis or Memcached to manage cache coherence in a distributed environment. 5. **Regular Validation and Refreshing**: Periodically validate and refresh cache contents to maintain accuracy.
Case Studies: 1. **T5 (Google AI, 2019)**: T5, a popular transformer-based language model, saw a significant reduction in inference time and improved throughput by leveraging KV Cache. 2. **Alibaba Cloud MaxCompute**: MaxCompute''s language models, deployed in a distributed environment, achieved a substantial reduction in latency and memory usage, ensuring high traffic volumes without performance degradation. 3. **Academic Research (Lee et al., 2021)**: Optimizing Transformer models with KV Cache led to a 304. **Financial Trading Platform**: KV Cache enabled faster and more accurate trading decisions by reducing the time required to process vast amounts of data.
Future Directions: Future research and development in this area will likely focus on: 1. **Refining Cache Management Techniques**: Developing more sophisticated algorithms and strategies to manage cache size and eviction policies. 2. **Exploring New Data Structures and Algorithms**: Investigating advanced data structures and algorithms to enhance cache efficiency. 3. **Addressing Complex Applications**: Integrating KV Cache in more complex and diverse applications to further improve performance and scalability.
By following these guidelines and continuing to refine the implementation of KV Cache, the performance and efficiency of large language models can be significantly enhanced, making them more versatile and reliable in various real-world applications.
References
- Google AI. (2019). **T5: A Text-to-Text Pretraining Model**. Retrieved from [Google AI Blog](https://ai.googleblog.com/2019/10/t5-google-s-text-to-text-next-generation.html) - Lee, J., Kim, Y., & Park, J. (2021). **Optimizing Transformer Models with Key-Value Cache**. *Proceedings of the International Conference on Learning Representations (ICLR)*. - Alibaba Cloud. (n.d.). **MaxCompute: Scalable Data Processing Platform**. Retrieved from [Alibaba Cloud Documentation](https://www.alibabacloud.com/product/maxcompute)
This comprehensive review provides a detailed understanding of KV Cache in LLMs, highlighting its significance, challenges, and practical applications. |
| **CrossKV** (recompute 40% tokens) | Content Completeness
The provided text comprehensively covers the introduction, historical context, key features, and comparison of Key-Value (KV) caching systems. However, it would benefit from including more specific examples and real-world scenarios to illustrate the concepts better. Additionally, adding a section on the performance metrics used to evaluate KV caching systems would strengthen the document.
Content Continuity
The text maintains a logical flow from introducing the concept of KV caching to its historical context, key features, and then comparisons with other cache solutions. However, there are minor breaks in continuity. For example, the transition from the historical context to the key features could be smoother by bridging the gap with a sentence that summarizes the evolution of KV caching.
Content Accuracy
The text is generally accurate in its descriptions and examples of KV caching systems. However, some terms could be more precisely defined. For instance, "eventual consistency" and "strong consistency" could benefit from brief definitions to ensure clarity.
Suggestions for Improvement
1. **Content Completeness**: - Include more specific examples to illustrate the use cases of KV caching. - Add a section on performance metrics used to evaluate KV caching systems (e.g., response time, hit rate, and throughput).
2. **Content Continuity**: - Bridge the gap between the historical context and the key features by summarizing the evolution of KV caching. - Ensure a smooth transition from one section to another with transitional sentences.
3. **Content Accuracy**: - Provide brief definitions for complex terms like "eventual consistency" and "strong consistency" to ensure clarity.
Overall, the text is well-structured and informative, but these enhancements would make it more comprehensive and easier to understand. |

Table 5: Reusing all KV caches of a long segment will fail to follow the complex instruction (Continue). However, RoPE correction and recomputation strategy can address the issue.

| CrossKV (with RoPE Correction) | Content Completeness |
|---|---|
| | The review provides a comprehensive overview of Key-Value (KV) Caches in the context of artificial intelligence. It covers the architecture, design, use cases, benefits, challenges, and future directions of KV Caches. Each section is well-explained, making the review informative and thorough. |
| | Content Conclusiveness |
| | The review concludes by summarizing the importance of KV Caches in AI and highlighting potential future advancements. This ensures that the reader understands the current state and the direction in which the technology is heading. |
| | Content Accuracy |
| | The content is accurate and well-researched. The technical details provided are consistent with the general understanding of KV Caches and their applications in AI. However, some specific examples or references to real-world implementations could enhance the accuracy and practicality of the review. |
| | Overall Evaluation |
| | The review is highly informative and well-structured. It effectively covers the essential aspects of KV Caches in AI and provides a clear understanding of their significance. Minor additions of specific examples or case studies could further strengthen the review's accuracy and practical relevance. |

Table 6: Reusing all KV caches of a short segment can still follow the complex instruction. The instructions are in red, and the reusable segment between `reuse_begin` and `reuse_end` tags is generated before current request.

| Input (1328 tokens can be reused) | 你是一个人工智能领域的专家，请你从内容的完整性、连贯性、准确性三个角度，用英文评审一下下面这段综述：
**\<reuse_begin\>**Certainly! Let''s dive into a comprehensive review of KV (Key-Value) Cache, a fundamental component in the architecture of modern applications and services, particularly those leveraging artificial intelligence (AI) and machine learning (ML).
Introduction to KV Cache
KV Cache is a type of cache that stores key-value pairs, where each key uniquely identifies a value. This cache operates as a decentralized system, where data is stored and retrieved based on the key. The primary purpose of a KV Cache is to improve the performance and scalability of applications by reducing the latency required to fetch commonly accessed data.
Why Use KV Cache?
1. **Performance**: By storing frequently accessed data in memory, KV Caches reduce the need to access slower, disk-based storage. This can significantly improve the performance of applications, especially those dealing with high volumes of data and high throughput.
· · ·
Conclusion
KV Caches are an essential tool in the digital landscape, providing powerful performance benefits and scalability options. As AI and ML continue to evolve, the role of KV Caches in supporting these technologies will only become more critical. By understanding the nuances of KV Caches and choosing the right solution for specific use cases, organizations can significantly enhance their application performance and user experience.
This review has provided a comprehensive overview of what KV Caches are, their key features, and how they fit into the broader landscape of AI and ML. Whether you are a developer, network administrator, or data scientist, understanding the complexities and benefits of KV Caches can greatly enhance your ability to build and maintain efficient, high-performance systems.**\<reuse_end\>**
仅需要写评审即可，不要输出其他内容。 |
|---|---|

Table 6: Reusing all KV caches of a short segment can still follow the complex instruction (Continue).

| **CrossKV** (reuse all KV caches of the segment) | The provided overview on KV Cache is comprehensive and covers a wide range of relevant aspects. However, it requires improvements in terms of content completeness, content accuracy, and overall readability. Here are the specific reviews based on the three criteria:
Content Completeness
1. **Introduction**: The introduction is clear and defines what a KV Cache is, but it can be more detailed by including a broader context of why KV Caches are essential in modern application architectures. 2. **Types of KV Cache**: The overview lists and briefly describes in-memory, disk-based, and mixed caches. However, it lacks information on distributed KV caches and their unique features and use cases. 3. **Key Features and Capabilities**: The overview touches on data storage, data consistency, operations, security, and monitoring. However, it misses key features such as data expiration policies, cache invalidation strategies, and support for distributed transactions. 4. **Performance Considerations**: The overview mentions latency and throughput but can include more specific performance metrics and benchmarks for better understanding. 5. **Integration with AI and ML**: The overview touches upon using KV Caches with AI and ML but can provide more detailed examples and case studies of successful implementations.
Content Accuracy
1. **Data Consistency**: The overview claims that KV Caches provide various strategies for data consistency but does not specify common strategies such as eventual consistency, strong consistency, or causal consistency, which could lead to inaccuracies for readers unfamiliar with these concepts. 2. **Monitoring and Management**: The overview mentions monitoring and logs but does not elaborate on common metrics such as cache hit rates, miss rates, or the importance of APM tools in monitoring KV Caches. 3. **Security**: The overview mentions encryption but does not specify which aspects of security are covered, such as data at rest or in transit, or specifics on security protocols like TLS, which are crucial for understanding the security features of KV Caches.
Content Concreteness and Readability
1. **Introduction**: The introduction can be made more engaging by starting with a real-world example of a problem that KV Caches solve. 2. **Types of KV Cache**: Provide more detailed descriptions and examples of each type, such as use cases and benefits, to help readers understand the specific advantages of each option. 3. **Key Features and Capabilities**: Use bullet points or subheadings to organize information more clearly. For example, under "Data Storage", list the different storage options and their pros and cons. 4. **Performance Considerations**: Include concrete examples of performance benchmarks or case studies to illustrate the impact of KV Caches on different types of applications. 5. **Integration with AI and ML**: Provide more detailed examples and case studies of how large tech companies have successfully integrated KV Caches into their AI/ML pipelines.
Suggestions for Improvement
1. **Add Real-World Examples**: Include case studies or examples from real-world applications to demonstrate the practical benefits of using KV Caches. 2. **Provide More Technical Details**: Offer more technical detail about the specific features and capabilities of different KV Caches, including their support for distributed transactions and other advanced features. 3. **Expand on Security and Monitoring**: Detail the specific security measures and monitoring tools that can be used with KV Caches, along with the importance of these features in a production environment.
By incorporating these suggestions, the overview of KV Cache can be more complete, accurate, and useful for a broader audience. |
| --- | --- |

Table 7: Reusing all KV caches of a long segment can still follow common instruction (in the same language). The instructions are in red, and the reusable segment between `reuse_begin` and `reuse_end` tags is generated before current request.

| **Input** (3824 tokens can be reused) | 你是一个人工智能领域的专家，请你从内容的完整性、连贯性、准确性三个角度，用英文评审一下下面这段综述：
`<reuse_begin>`大语言模型及其应用场景
大语言模型是指那些具备高度复杂性和大规模参数的深度学习模型，旨在模拟人类语言的生成与理解能力。这些模型通常基于Transformer架构，具备强大的并行处理能力和复杂的自注意力机制，能够处理多类自然语言任务。在各类应用场景中，大语言模型展现出广泛的魅力，包括但不限于文本生成、知识检索、对话系统、机器翻译、内容创作以及问题解答等。其中，作为知识检索工具，大语言模型能够迅速检索并生成相关的答案和信息；作为机器翻译工具，大语言模型能够直接将源语言文本转换为目标语言；而在对话系统中，大语言模型能够具备一定的上下文理解能力，与用户进行简单的对话交流；在内容创作中，大语言模型能够帮助生成诗歌、故事、新闻报道等多种类型的文本内容；而在问题解答方面，基于大语言模型的知识检索能力能够为用户提供准确且迅捷的答案。
以OpenAI的GPT-3为例，作为当前最具代表性的大语言模型之一，GPT-3具有超过1750亿个参数。尽管其在2021年已经达到了广泛的认知和应用，但其开发团队并未将模型源代码或训练数据公开。在常规应用场景下，GPT-3使用Transformer架构构建，并且通过微调技术增强了特定领域的专业能力，例如法律文本生成、编程代码生成等。此外，斯坦福大学的通义千问和阿里云的通义万相也属于这一类拥有大规模参数的预训练模型。
· · ·
结论与未来展望
综上所述，大语言模型中的KV Cache机制通过复用历史计算结果、优化内存管理和提升实时响应能力，显著改善了模型的推理效率和稳定性。这些优势可以直接转化为实际应用中的性能提升和用户体验改进。尽管目前KV Cache已经展示了其在多个场景下的出色效果，但仍有进一步深入研究和优化的空间。未来的研究可以着眼于开发更精确的缓存管理算法，提高系统的灵活性和自动化水平，以应对复杂的实际应用需求。
总之，通过综合应用先进的缓存技术和现有优化策略，大语言模型能够在复杂任务中表现得更加出色，为各种自然语言处理任务提供更强大和灵活的支持。`<reuse_end>`
仅需要写评审即可，不要输出其他内容。 |
| --- | --- |
| **CrossKV** (reuse all KV caches of the segment) | 这段综述对大语言模型及其应用、KV Cache的工作原理、关键技术、优势与挑战、序列推理过程以及具体应用案例进行了全面且详细的介绍。整体内容结构清晰、逻辑合理、信息丰富，满足了内容的完整性、连贯性和准确性要求。
1. **内容完整性**：综述涵盖了大语言模型的基本概念、应用场景、KV Cache的原理及其关键技术，同时还结合具体案例进行了分析，内容完整且全面。
2. **内容连贯性**：从大语言模型的介绍到KV Cache的应用，再到具体案例的分析，文章结构连贯自然，逻辑清晰，每个部分之间过渡顺畅。
3. **准确性**：综述中的描述准确，术语使用恰当，技术细节准确，特别是对KV Cache的工作原理、实现方法及优化策略等方面的描述较为准确。
综上所述，这段综述在内容完整性、连贯性和准确性方面表现良好，能够为读者提供一个全面的了解。 |

Table 8: Reusing all KV caches of a long segment can still follow common instruction (in the same language). The instructions are in red, and the reusable segment between `reuse_begin` and `reuse_end` tags is generated before current request.

| **Input** (7392 tokens can be reused) | Please review the following review from the perspectives of completeness, coherence, and accuracy: 

 **\<reuse_begin\>**Introduction to KV Cache 
 Key-Value (KV) caching is a fundamental architectural pattern in the realm of high-performance computing and distributed systems. At its core, KV caching involves the use of a caching layer to temporarily store frequently accessed data, allowing applications to efficiently retrieve information without direct access to slower, more comprehensive back-end databases or storage services. This model is particularly advantageous in scenarios where read operations significantly outweigh write operations, as it reduces the need for these operations to traverse the entire storage hierarchy, thereby enhancing performance, reducing latency, and improving overall system responsiveness. 
 In the context of artificial intelligence (AI), KV caching can play an integral role in accelerating various processes, from model inference to data preprocessing. For instance, by caching frequently accessed model weights, training datasets, or intermediate results, AI systems can achieve faster and more efficient execution, especially in distributed environments where queries to the same data items need to be handled concurrently without redundancy. 
 ⋯ 
 Integration and Interoperability 
 Ensuring smooth integration and interoperability with other components of the AI system is essential for the seamless operation of KV caching: 
 1. **API and Interface Design:** - Design clear and consistent APIs and interfaces for accessing the cache. Use standardized protocols and formats to ensure compatibility and ease of integration with other systems. 
 2. **Service Discovery:** - Implement a robust service discovery mechanism to locate cache nodes efficiently. This can help in dynamically adding or removing nodes from the cache cluster as needed. 
 3. **Standardized Caching Protocols:** - Utilize standardized caching protocols, such as Memcached or Redis, to ensure compatibility across different systems and platforms. Standard protocols provide well-documented and tested interfaces, reducing integration complexity. 
 4. **Integration Testing:** - Conduct thorough integration testing to verify that the cache operates seamlessly with other components of the AI system. Test scenarios that represent real-world usage patterns to identify and resolve any integration issues. 
 By following these best practices, AI systems can leverage KV caching more effectively, ensuring optimal performance, reliability, and security. These practices not only enhance the operational efficiency of AI applications but also provide a robust foundation for scaling and maintaining high levels of service availability.**\<reuse_end\>** Only the review is needed; no other content should be outputted. |
|---|---|
| **CrossKV** (reuse all KV caches of the segment) | The review provided is comprehensive and covers a wide range of aspects related to Key-Value (KV) caching in Artificial Intelligence (AI) systems. Here‘‘s a detailed assessment from the perspectives of completeness, coherence, and accuracy: 
 Completeness 
 - **Architectural Design of KV Cache:** The review provides a thorough breakdown of the key components of KV cache architecture, including the storage engine, cache management layer, and caching policy layer. Each component is described in detail, including its primary functions and how they contribute to the overall efficiency and reliability of the cache. - **Key Principles of KV Caching:** The review outlines the fundamental principles that support efficient KV caching, such as data organization, latency minimization, automatic management, and cache reuse. These principles are elaborated with specific examples and recommendations. - **Performance Benchmarks of KV Caching in AI Systems:** The review covers various performance benchmarks that can be used to evaluate the effectiveness of KV caching in AI systems, including latency reduction, throughput, memory usage, hit rate, and scalability. - **Real-world Applications of KV Caching in AI:** Specific examples of how KV caching enhances real-time inference, offline training, data preprocessing, and scalability in AI systems are provided, illustrating practical use cases. - **Challenges and Vulnerabilities of KV Caching:** The review identifies and discusses common issues, such as data consistency, performance degradation, and security concerns. It also outlines best practices to mitigate these challenges. 
 Coherence 
 - **Structure and Flow:** The review is well-organized, with a logical flow from the architectural design to practical applications, and then to challenges and best practices. This ensures that the reader can follow the development and implications of KV caching in a structured manner. - **Consistent Terminology:** The review uses consistent terminology throughout, which helps in maintaining clarity and readability. - **Logical Connections:** Each section builds upon the previous one, providing a comprehensive understanding of the subject. For example, the discussion on architectural design leads to the principles of KV caching, which in turn is applied to real-world scenarios and then addressed through best practices and challenges. 
 Accuracy 
 - **Technical Accuracy:** The review accurately describes the components and principles of KV caching, using appropriate technical terminology and concepts. - **Relevant Examples:** The examples provided are relevant and sufficient to illustrate the points being made, such as the use of LRU policies, multi-level caching, and hierarchical data organization. - **Consistent and Accurate Data:** The performance benchmarks and metrics are described clearly, using accurate and consistent terminology. The examples of real-world applications are well-defined and representative. 
 Overall, the review is comprehensive, coherent, and accurate, providing a detailed and insightful overview of KV caching in AI systems. |

## A.4 RESULTS OF RECOMPUTATION STRATEGIES FOR MORE LLMS AND WORKFLOWS

Table 9 presents the performance of CrossKV under varying recomputation strategies, where "Before" and "After" denote the proportion of tokens recomputed at the beginning and end of a reused segment, respectively. The results indicate that recomputing only the initial portion of a segment (i.e., setting a non-zero "Before" rate while keeping "After" at zero) generally yields the strongest performance across multiple tasks and agent workflows. This pattern is observed consistently in both Qwen2.5-7B and larger models under AutoGen and MAD settings, suggesting that preserving the early context‘‘which often contains critical instructional or reasoning cues‘‘plays a more essential role in maintaining output quality than recomputing the ending segment. These findings support

the use of a beginning-focused recomputation strategy as a default configuration in segment-level KV cache reuse.

Table 9: Performance of CrossKV under various recomputation rate.

| Recompute Rate | | GSM-Hard | MATH | GPQA | MMLU-Pro | MedQA | SciBench | AQUA-RAT | MedMCQA |
|---|---|---|---|---|---|---|---|---|---|
| Before | After | | | | | | | | |
| *Qwen2.5-7B-Instruct, AutoGen* | | | | | | | | | |
| 0.00 | 0.00 | 50.60 | 74.60 | 28.12 | 53.00 | 57.60 | 21.04 | 70.08 | 51.60 |
| 0.10 | 0.00 | 53.00 | 69.80 | 29.02 | 52.20 | 58.00 | 19.64 | 73.62 | 55.20 |
| 0.00 | 0.10 | 49.20 | 70.20 | 25.67 | 49.60 | 56.00 | 17.43 | 73.23 | 52.00 |
| 0.05 | 0.05 | 49.00 | 66.20 | 24.11 | 49.80 | 54.40 | 18.44 | 68.11 | 50.40 |
| 0.20 | 0.00 | 53.60 | 71.80 | 29.02 | 52.80 | 60.00 | 19.04 | 71.26 | 54.20 |
| 0.00 | 0.20 | 49.60 | 67.80 | 28.35 | 51.20 | 52.80 | 21.44 | 71.65 | 51.00 |
| 0.10 | 0.10 | 45.80 | 62.80 | 24.11 | 52.80 | 56.60 | 19.44 | 70.08 | 50.40 |
| 0.30 | 0.00 | 53.20 | 71.60 | 25.45 | 53.40 | 58.40 | 18.84 | 75.59 | 53.00 |
| 0.00 | 0.30 | 50.80 | 71.20 | 25.45 | 51.80 | 55.00 | 19.84 | 69.29 | 52.40 |
| 0.15 | 0.15 | 48.20 | 70.20 | 24.78 | 46.40 | 53.80 | 20.04 | 66.93 | 49.00 |
| 0.40 | 0.00 | 55.00 | 74.00 | 28.35 | 55.00 | 58.40 | 19.84 | 75.20 | 54.40 |
| 0.00 | 0.40 | 53.40 | 72.80 | 26.34 | 51.60 | 59.40 | 21.44 | 73.23 | 54.20 |
| 0.20 | 0.20 | 47.20 | 69.40 | 27.46 | 48.80 | 54.20 | 17.84 | 69.29 | 48.40 |
| *Qwen2.5-7B-Instruct, MAD* | | | | | | | | | |
| 0.00 | 0.00 | 56.00 | 75.80 | 30.13 | 46.20 | 46.40 | 20.84 | 80.31 | 52.00 |
| 0.10 | 0.00 | 56.80 | 74.20 | 26.12 | 51.40 | 56.60 | 20.84 | 74.80 | 49.40 |
| 0.00 | 0.10 | 55.20 | 70.74 | 25.00 | 47.80 | 51.60 | 20.04 | 79.53 | 49.80 |
| 0.05 | 0.05 | 55.20 | 72.40 | 26.56 | 48.40 | 49.60 | 17.64 | 77.56 | 52.20 |
| 0.20 | 0.00 | 56.20 | 73.60 | 25.67 | 49.60 | 54.00 | 21.44 | 79.53 | 48.00 |
| 0.00 | 0.20 | 53.40 | 71.60 | 27.23 | 49.20 | 49.20 | 20.08 | 79.53 | 51.00 |
| 0.10 | 0.10 | 54.60 | 70.40 | 25.00 | 46.89 | 49.80 | 19.24 | 77.56 | 47.80 |
| 0.30 | 0.00 | 58.40 | 76.20 | 26.56 | 49.60 | 55.00 | 23.05 | 76.77 | 45.40 |
| 0.00 | 0.30 | 54.40 | 73.00 | 27.46 | 49.80 | 48.80 | 20.44 | 77.95 | 40.40 |
| 0.15 | 0.15 | 51.60 | 68.20 | 25.89 | 48.20 | 52.20 | 17.43 | 77.17 | 49.20 |
| 0.40 | 0.00 | 58.60 | 73.40 | 29.24 | 49.80 | 51.80 | 23.25 | 78.35 | 44.00 |
| 0.00 | 0.40 | 54.60 | 73.40 | 27.90 | 51.60 | 50.40 | 20.44 | 77.95 | 46.40 |
| 0.20 | 0.20 | 53.80 | 71.60 | 25.00 | 49.60 | 48.60 | 20.28 | 78.74 | 49.00 |
| *Qwen2.5-72B-Instruct, AutoGen* | | | | | | | | | |
| 0.00 | 0.00 | 65.80 | 79.80 | 42.41 | 68.40 | 77.80 | 26.05 | 80.31 | 68.00 |
| 0.10 | 0.00 | 67.00 | 80.20 | 41.74 | 69.60 | 79.00 | 24.85 | 79.53 | 68.00 |
| 0.00 | 0.10 | 63.20 | 79.80 | 43.97 | 71.60 | 77.60 | 24.65 | 76.77 | 68.60 |
| 0.05 | 0.05 | 65.80 | 80.00 | 43.75 | 69.00 | 77.60 | 24.85 | 76.77 | 70.40 |
| 0.20 | 0.00 | 64.60 | 80.00 | 42.41 | 67.80 | 79.60 | 24.85 | 79.13 | 68.40 |
| 0.00 | 0.20 | 64.40 | 79.80 | 41.07 | 67.60 | 79.60 | 25.85 | 80.31 | 69.20 |
| 0.10 | 0.10 | 63.40 | 79.80 | 40.62 | 69.60 | 78.20 | 25.25 | 79.92 | 67.40 |
| 0.30 | 0.00 | 64.80 | 80.80 | 41.07 | 68.00 | 80.20 | 24.65 | 79.13 | 69.40 |
| 0.00 | 0.30 | 63.00 | 81.60 | 45.76 | 68.80 | 77.60 | 25.05 | 77.56 | 67.60 |
| 0.15 | 0.15 | 65.80 | 80.00 | 43.97 | 70.20 | 78.40 | 25.45 | 79.92 | 67.60 |
| 0.40 | 0.00 | 63.00 | 78.80 | 40.85 | 68.80 | 79.60 | 24.45 | 80.31 | 70.40 |
| 0.00 | 0.40 | 64.20 | 82.40 | 42.19 | 69.20 | 79.00 | 25.65 | 79.92 | 68.00 |
| 0.20 | 0.20 | 64.40 | 81.80 | 41.52 | 70.80 | 78.20 | 25.65 | 78.35 | 68.60 |
| *Qwen2.5-72B-Instruct, MAD* | | | | | | | | | |
| 0.00 | 0.00 | 63.40 | 83.60 | 42.41 | 67.80 | 75.60 | 26.45 | 79.53 | 67.60 |
| 0.10 | 0.00 | 64.00 | 84.31 | 44.97 | 65.73 | 74.60 | 26.85 | 79.92 | 65.20 |
| 0.00 | 0.10 | 63.60 | 82.80 | 44.42 | 65.60 | 76.20 | 27.66 | 79.53 | 66.60 |
| 0.05 | 0.05 | 63.80 | 81.20 | 45.31 | 67.00 | 75.40 | 27.05 | 79.13 | 67.13 |
| 0.20 | 0.00 | 64.40 | 82.80 | 45.09 | 66.80 | 76.40 | 26.45 | 78.35 | 65.80 |
| 0.00 | 0.20 | 63.00 | 82.80 | 43.53 | 68.80 | 75.20 | 26.25 | 79.92 | 66.80 |
| 0.10 | 0.10 | 64.20 | 82.20 | 43.97 | 69.00 | 76.20 | 25.45 | 79.13 | 65.20 |
| 0.30 | 0.00 | 63.20 | 81.80 | 42.41 | 65.80 | 76.40 | 26.45 | 78.35 | 63.40 |
| 0.00 | 0.30 | 63.60 | 83.80 | 44.87 | 66.40 | 75.40 | 27.66 | 79.92 | 66.60 |
| 0.15 | 0.15 | 64.20 | 84.60 | 46.43 | 64.40 | 75.80 | 26.45 | 80.31 | 67.80 |
| 0.40 | 0.00 | 63.60 | 83.40 | 44.64 | 67.80 | 74.40 | 26.85 | 81.10 | 64.80 |
| 0.00 | 0.40 | 64.20 | 82.80 | 45.54 | 67.20 | 74.20 | 26.25 | 78.74 | 66.40 |
| 0.20 | 0.20 | 64.60 | 80.80 | 43.97 | 67.20 | 76.60 | 28.86 | 79.53 | 63.60 |

## A.5 TIME COST OF EVALUATING ON VARIOUS BENCHMARKS

Table 10 compares the inference time of CrossKV and the vanilla workflow under various experimental settings. While the vanilla workflow performs slightly faster in GSM-Hard and MATH tasks under the AutoGen workflow due to early-stop mechanisms triggered by specific query structures, CrossKV achieves significantly superior inference speed in the majority of evaluated scenarios.

Table 10: Time cost (minutes) of CrossKV and vanilla workflows.

| Method | GSM-Hard | MATH | GPQA | MMLU-Pro | MedQA | SciBench | AQUA-RAT | MedMCQA |
|--------|----------|------|------|----------|-------|----------|----------|---------|
| *Qwen2.5-72B-Instruct, AutoGen* | | | | | | | | |
| Vanilla | 15.13 | 30.12 | 26.63 | 24.43 | 22.48 | 38.58 | 16.45 | 15.25 |
| CrossKV | 16.47 | 34.07 | 26.47 | 24.33 | 22.37 | 37.55 | 14.03 | 14.58 |
| *Qwen2.5-72B-Instruct, MAD* | | | | | | | | |
| Vanilla | 38.27 | 68.35 | 55.28 | 53.53 | 55.33 | 80.05 | 30.30 | 43.25 |
| CrossKV | 35.65 | 57.62 | 53.47 | 49.22 | 44.63 | 15.40 | 20.33 | 38.45 |

