# OpenReview forum: "Towards a Collaborative Memory for Agentic Workflow: Breaking the Prefix Barrier with Segment-Level KV Cache Sharing"
_ICLR.cc/2026/Conference — Submitted to ICLR 2026_

### Official Review · Reviewer_taJt · 2025-10-29

**Soundness:** 3
**Presentation:** 2
**Contribution:** 3
**Rating:** 6
**Confidence:** 3

**Summary:**

This paper addresses the inefficiency of KV cache reuse in multi-agent LLM systems. Existing methods rely on strict prefix matching, making cache reuse rare under heterogeneous prompts. The authors propose a Segment-Level KV Cache Sharing mechanism that decomposes the cache into semantically coherent segments, enabling agents to reuse KV segments across contexts without prefix alignment.
They implement a high-performance prototype, CrossKV, on top of the vLLM engine using PageAttention, which introduces a memory table for segment-level KV aliasing and retrieval.
Extensive experiments on multiple models (Qwen2.5, Llama3) and agentic workflows (AutoGen, MAD, Solver) demonstrate up to 4.6× TTFT speedup and even performance gains in several reasoning benchmarks. The paper also analyzes the effect of positional encoding (RoPE) and proposes adaptive partial recomputation strategies.

**Strengths:**

1. Segment-level KV cache sharing breaks the prefix-matching bottleneck in LLM inference.
2. CrossKV is a complete, model-agnostic prototype built on vLLM.
3. Includes in-depth discussion on positional encoding, recomputation, and memory overhead.

**Weaknesses:**

1. Lack of formal theory explaining semantic stability of reused KV segments.
2. Limited comparison to other advanced caching methods (e.g., KVShare, CacheBlend).
3. RoPE correction introduces extra memory and computation overhead, and long-segment reuse may require partial recomputation, increasing system complexity.
4. Absence of large-scale real-world multi-agent case studies.

**Questions:**

1. How are the semantic boundaries of segments detected or defined in practice?
2. Could segment aliasing introduce semantic drift or context leakage across agents?
3. How would CrossKV behave in cross-lingual or high-noise agent communication?
4. Is there any safeguard to prevent incorrect segment matching (hash collision or semantic mismatch)?
5. Could this approach be integrated with retrieval-augmented or external-memory systems?

---

> ### Author Response · Authors · 2025-11-20
> **Response to Reviewer taJt [1/3]**
>
> **Q1: Lack of formal theory explaining semantic stability of reused KV segments.**
>
> **A1:**  We thank the reviewer for raising this important point. We agree that a deeper theoretical justification is valuable, and we are pleased to provide a formal equivalence proof that solidifies the theoretical foundation of our method.
>
> Specifically, with RoPE correction, KV cache reuse is theoretically equivalent to computing attention under a particular, semantically meaningful mask. To illustrate this, consider the following setting:
> - Let $Q_A$ and $O_A$ denote the input query and output of Agent-A
> - Let $Q_B = [Q_{B1}, O_A, Q_{B2}]$ denotes the input query of Agent-B, which incorporates Agent-A's output, and $O_B$ denotes the output of Agent-B.
>
> The attention mask configuration reveals the fundamental difference between the two approaches:
>
> **1. Vanilla Attention Mask (without KV Cache Reuse):**
>
> | Q \ K  | Q_A |  Q_B1 | O_A | Q_B2 | O_B |
> | ------ | --- | ---- | --- | ---- | --- |
> | Q_{B1} | 0 | 1 | 0| 0 | 0 |
> | O_A    | 0   |  1    | 1   | 0    | 0   |
> | Q_{B2} | 0   |  1    | 1   | 1    | 0   |
> | O_B    | 0   |  1    | 1   | 1    | 1   |
>
> In this configuration, Agent-B must re-interpret $O_A$ without access to its original generative context ($Q_A$), potentially leading to semantic ambiguity.
>
> **2. CrossKV Attention Mask (with KV Cache Reuse):**
>
> | Q \ K  | Q_A |  Q_B1 | O_A | Q_B2 | O_B |
> | ------ | --- | ---- | --- | ---- | --- |
> | Q_{B1} | 0   | 1    | 0   | 0    | 0   |
> | O_A    | 1   | 0    | 1   | 0    | 0   |
> | Q_{B2} | 0   | 1    | 1   | 1    | 0   |
> | O_B    | 0   | 1    | 1   | 1    | 1   |
>
> The critical distinction is that CrossKV preserves $O_A$'s connection to its original context ($Q_A$), allowing Agent-B to understand $O_A$ from the same perspective as Agent-A generated it. This maintains semantic continuity while providing computational efficiency. Given that $O_A = \text{LLM}([Q_A])$ is solely a function of its immediate context $Q_A$, it follows that its attentional grounding remains there, correctly excluding $Q_{B1}$ from its attention map.
>
> Such a pattern of attention mask represents the inherent collaboration in MAS, where agents share a common goal and context. For example, in a Critic workflow setup, the Critic's prompt is often: "The Actor's response is: {Output_Actor}. Please critique it." Reusing the Actor's KV cache for Output_Actor allows the Critic to critique the output **from the same contextual perspective as the Actor**, leading to more coherent and grounded evaluations. This is not only more efficient but also **theoretically more aligned** with the goal of maintaining semantic continuity across agents.
>
> ---
> **Q2: Limited comparison to other advanced caching methods (e.g., KVShare, CacheBlend).**
>
> **A2:** We thank the reviewer for this pertinent question regarding baseline comparisons. While our core contribution is for MAS, our method is generalizable. To directly address the reviewer's point, we conduct experiments in a RAG setting.
> Following the setup of KVShare, we use the Qwen2.5-7B-Instruct and evaluate on the SAMSum dataset. Notably, our reproduced vanilla baseline (without KV cache reuse) yields significantly higher RougeL than that reported in the original KVShare study (38.99 vs. 20.17). Even when compared against this stronger baseline, CrossKV nearly matches its performance (38.88), effectively maintaining output quality while enabling efficient KV reuse.
>
> | Method           | Vanilla (KVShare) | CacheBlend@0.1 | CacheBlend@0.4  | KVShare@0.1 | KVShare@0.4 | Vanilla (Ours) | CrossKV |
> | :--------------- | :---------------- | :------------- | :------------- | :------- | :---------- | :------------- | :------ |
> | **Rouge-L**       | 20.17             | 13.92 (-6.25)          | 15.80 (-4.37)          | 15.75 (-4.42)       | 17.21 (-2.96)       | 38.99          | 38.88 (-0.11)   |
>
> Nevertheless, the distinctions between these methods and our CrossKV are clarified in the following comparison.
>
> **1. Divergence in Problem Setting and Scope:** CacheBlend and KVShare are primarily designed for RAG, which necessitate a global KV cache pool shared across many unrelated queries. In contrast, our work targets Multi-Agent Systems, where we maintain a local, session-specific KV pool dedicated to a single task's workflow. MAS setting is a more natural fit for exact KV reuse, as the context (e.g., an agent's output) is generated and consumed within the same session, eliminating the need for the costly upfront pre-computation (prefill) required for external RAG documents.
>
> **2. Fundamental Methodological Differences:**
> - CacheBlend and KVShare introduces extra overhead for token selection, leading to a computational complexity. Moreover, KVShare employs a similarity-based reuse, where the KV cache of a similar but not identical context is retrieved.
> - Our method is based on exact-match reuse of verbatim token sequences and has no extra computation costs for token selection.

---

> ### Author Response · Authors · 2025-11-20
> **Response to Reviewer taJt [2/3]**
>
> **Q3:RoPE correction introduces extra memory and computation overhead, and long-segment reuse may require partial recomputation, increasing system complexity.**
>
> **A3:** We appreciate the reviewer's comment on overhead. Let us clarify the computational comparison:
>
> **RoPE Correction Overhead:** Involves a minimal, linear-time operation to apply positional rotations to the pre-existing keys in the cache. Also the GPU memory can also be saved by the KV cache eviction mechanism in the vLLM platform.
>
> **Partial Recomputation Overhead:** Involves a minimal, linear-time operation to apply positional rotations to the pre-existing keys in the cache.
>
> **Full KV Recomputation Cost:** Requires a full forward pass of the entire segment through the model's embedding, attention, and MLP layers, which is quadratic in sequence length and demands considerable memory bandwidth.
>
> The former is a negligible cost compared to the latter. Our presented TTFT (time to first token) metrics in the table below quantitatively confirm that the net efficiency gain of CrossKV, even with these mechanisms, is robust and positive.
> | \#Reused tokens | TTFT of vanilla | TTFT of CrossKV | TTFT of CrossKV+RoPE correction |
> | --------------- | --------------- | --------------- | ------------------------------- |
> | 0.5k            | 130.34 ms       | 92.73 ms        | 105.33 ms                       |
> | 2k              | 278.28 ms       | 113.25 ms       | 128.07 ms                       |
> | 8k              | 818.89 ms       | 114.76 ms       | 131.86 ms                                |
>
> ---
> **Q4: Absence of large-scale real-world multi-agent case studies.**
>
> **A4:** We thank the reviewer for this valuable feedback regarding the evaluation scale. We agree that validation on complex, large-scale tasks is crucial. To directly address this point and demonstrate the effectiveness of our method in a more challenging and realistic setting, we have conducted additional experiments on the GAIA (Text) benchmark, which is designed to mimic real-world question-answering tasks requiring multi-step reasoning.
>
> In this experiment, conducted using the Qwen2.5-72B-Instruct model within the X-Master multi-agent workflow, we demonstrate that CrossKV maintains functional correctness while delivering enhanced performance.
>
> | Method  | Accuracy (GAIA) |
> |:------- |:--------------- |
> | Vanilla | 13.5%           |
> | CrossKV | **15.5%**       |
>
> This experiment serves as a compelling case study, demonstrating that CrossKV successfully accelerates a large-scale, multi-step reasoning system without degrading—and in this case, even improving—the outcome quality. We view this as strong preliminary evidence of its applicability to real-world scenarios. We will include these results in the revised manuscript and will continue to pursue more extensive large-scale evaluations as a key direction for future work.
>
> ---
> **Q5: How are the semantic boundaries of segments detected or defined in practice?
> Could segment aliasing introduce semantic drift or context leakage across agents?**
>
> **A5:** We appreciate the reviewer's concerns about semantic integrity. We provide firm answers below:
>
> **Segment Definition:** Segments can be explicitly defined by the multi-agent workflow. When the framework routes Agent-A's output $O_A$ to Agent-B, it deterministically wraps $O_A$ with 〈reuse\_begin〉 and 〈reuse\_end〉tags. This makes segment boundaries precise and unambiguous.
>
> **Semantic Drift & Leakage:** The reuse of an identical segment's KV cache is a form of exact computation reuse. It does not alter the semantic content of the segment (no drift). Moreover, since the KV cache is a deterministic function of a specific input sequence, it cannot contain or "leak" information from unrelated contexts. What is shared is the intended, common context between collaborative agents, which is by design and actually prevents the ambiguity that could arise from re-computation.

---

> ### Author Response · Authors · 2025-11-20
> **Response to Reviewer taJt [3/3]**
>
> **Q6: How would CrossKV behave in cross-lingual or high-noise agent communication?**
>
> **A6:** We thank the reviewer for raising these important edge cases regarding CrossKV's behavior in challenging scenarios.
>
> **Cross-lingual communication:** Appendix Table 4 illustrates that while naive long-segment reuse can struggle with complex, cross-lingual instructions due to positional misalignment, our mitigation strategies—RoPE correction and adaptive partial recomputation—effectively resolve this issue. These mechanisms ensure positional encodings remain consistent, allowing CrossKV to handle cross-lingual contexts robustly.
>
> **High-noise agent communication:** It is important to clarify that the noise would originate from the content of the inter-agent messages themselves, not from the KV cache sharing mechanism. CrossKV is a transparent optimization that faithfully reuses the computed representation of the input it is given. It does not amplify or inject noise; if the input context is noisy, the reused KV cache will reflect that noisy context accurately, just as a full recomputation would. Therefore, the presence and impact of noise are determined at the semantic, system-design level, and are orthogonal to the KV reuse process.
>
> ---
> **Q7: Is there any safeguard to prevent incorrect segment matching (hash collision or semantic mismatch)?**
>
> **A7:** Thank you for this important technical question.
> Our system employs an exact-match mechanism for segment reuse. A segment's KV cache is only reused when its content is a verbatim, byte-for-byte identical match to a previously computed segment. This strict requirement inherently prevents semantic mismatches, as even minor textual variations will trigger a full recomputation.
>
> This exact-match scenario is not a limitation but rather the predominant and most relevant case in multi-agent systems (MAS). In standard agent architectures, the output of one agent is passed precisely and completely as input to another. This design naturally creates numerous opportunities for exact segment matching, making our approach both robust and widely applicable in practice.
>
> ---
> **Q8: Could this approach be integrated with retrieval-augmented or external-memory systems?**
>
> **A8:** We thank the reviewer for this excellent question. CrossKV can indeed be integrated with and substantially accelerate retrieval-augmented systems.
>
> The core insight is that RAG workflows often involve verbatim reuse of retrieved context. CrossKV is designed precisely for this pattern. By building a cache of KV states for frequently accessed document segments, we can transform the standard "retrieve-then-prefill" pipeline into a much faster "retrieve-then-reuse" process. The results from our RAG experiments demonstrate this effective integration, where we observe a compelling performance profile of reduced latency without sacrificing accuracy.
>
> For a comparative analysis with CacheBlend and KVShare, please refer to our response in **A2**. To further validate CrossKV's applicability in RAG scenarios, we conduct additional evaluations using the Qwen3 model series on the musique-s dataset. The F1-score results are presented below:
>
> | Model                  | Vanilla (F1-score) | CrossKV (F1-score) | $\Delta$ (F1-score) | Vanilla (TTFT/s) | CrossKV (TTFT/s) | $\Delta$ (TTFT\%) |
> |:---------------------- |:------------------ |:------------------ | ------------------- | ---------------- | ---------------- | ----------------- |
> | Qwen3-8B-Instruct      | 0.3543             | 0.3548             | +0.0005             | 0.5893           | 0.1879           | -68.11\%          |
> | Qwen3-30B-A3B-Instruct | 0.4049             | 0.3856             | -0.0193             | 0.4018           | 0.1624           | -59.58%           |
> | Qwen3-32B-Instruct     | 0.3905             | 0.4069             | +0.0164             | 1.6956           | 0.2797           |  -83.50%                 |
>
> The results indicate that CrossKV maintains competitive performance (F1-score) across different model sizes while providing computational savings. The minor variation in scores falls within expectations for inference-time optimizations and demonstrates that our method effectively balances efficiency and accuracy.

---

> ### Author Response · Authors · 2025-11-27
> **Response to Reviewer taJt**
>
> We sincerely appreciate your time and expertise in reviewing our manuscript. As we have not yet received your feedback, we wanted to kindly inquire whether you might have any preliminary comments or questions that we could address at this stage. Your insights would be greatly valued in helping us further refine and strengthen our work.
>
> We truly hope that our response will contribute positively to the overall quality and impact of the paper, and we would be deeply grateful for any constructive feedback that could also support a favorable evaluation. Thank you once again for your thoughtful consideration and invaluable contribution to the peer-review process.

---

### Official Review · Reviewer_Ge4H · 2025-10-31

**Soundness:** 2
**Presentation:** 2
**Contribution:** 2
**Rating:** 2
**Confidence:** 3

**Summary:**

The paper proposes a method to address a key inefficiency in LLM-based multi-agent systems (MAS): the reliance of KV cache reuse systems on rigid prefix matching. This mechanism fails in MAS environments where agents have diverse prompt templates and contexts, leading to rare cache hits and significant redundant computation.

To solve this, the authors propose a Segment-Level KV Cache Sharing mechanism. This approach decomposes the KV cache into fine-grained "semantic segments". It allows any agent to reuse a cached segment from any other agent, regardless of its position in the new query, thereby enabling a collaborative working memory.

The paper also investigates the critical technical challenge of positional encoding (RoPE) mismatches and proposes an adaptive recomputation strategy as a solution. Experiments show the method significantly increases inference speed and, in some cases, improves task performance by enabling this working memory sharing.

**Strengths:**

* The paper correctly identifies a highly relevant and practical problem. As multi-agent systems become more common, the limitations of prefix-only caching become a critical bottleneck. The idea of a "collaborative memory" is a strong conceptual framing.
* The proposed mechanism has the intended effect of improving the cache hit rates and prefill speeds.

**Weaknesses:**

* The paper itself notes a lack of theoretical justification as a limitation. The fundamental assumption that a segment's KV cache is locally concentrated and largely context-independent is justified by citing sparsity literature and a single visual analysis (Fig. 1), but it's not deeply explored.
* The results are presented only with dense architectures, not with other more prevalent architectures like MoE. In general, such a technique, without theoretical justification, is hard to take for granted that it will generalize broadly.
* It is also not clear how the semantic segments are identified in practice. The LLM identifying reusable parts of its context upon prompting seems brittle.
* In the Table 1, it is not clear why certain results will actually improve upon reusing the cached KVs. Reusing cached KVs should strictly be a inference time win, not qualitative win.
* The manuscript in its current form is quite repetitive. The problem, solution, and contributions are restated in similar terms across the Abstract, Introduction, and Conclusion, diluting the paper's impact. The paper could be substantially shorter without losing any of its core technical merit.

**Questions:**

None

---

> ### Author Response · Authors · 2025-11-20
> **Response to Reviewer Ge4H [1/2]**
>
> **Q1: The paper itself notes a lack of theoretical justification as a limitation.**
>
> **A1:** We thank the reviewer for raising this important point. We agree that a deeper theoretical justification is valuable, and we are pleased to provide a formal equivalence analysis that solidifies the theoretical foundation of our method.
>
> Specifically, with RoPE correction, KV cache reuse is formally equivalent to computing attention under a particular, semantically meaningful mask. To illustrate this, consider the following setting:
> - Let $Q_A$ and $O_A$ denote the input query and output of Agent-A
> - Let $Q_B = [Q_{B1}, O_A, Q_{B2}]$ denotes the input query of Agent-B, which incorporates Agent-A's output, and $O_B$ denotes the output of Agent-B.
>
> The attention mask configuration reveals the fundamental difference between the two approaches:
>
> **1. Vanilla Attention Mask (without KV Cache Reuse):**
>
> | Q \ K  | Q_A | Q_B1 | O_A | Q_B2 | O_B |
> | ------ | --- | --- |  --- | ---- | --- |
> | Q_{B1} | 0   |  1    | 0   | 0    | 0   |
> | O_A    | 0   | 1    | 1   | 0    | 0   |
> | Q_{B2} | 0   | 1    | 1   | 1    | 0   |
> | O_B    | 0   | 1    | 1   | 1    | 1   |
>
> In this configuration, Agent-B must re-interpret $O_A$ without access to its original generative context ($Q_A$), potentially leading to semantic ambiguity.
>
> **2. CrossKV Attention Mask (with KV Cache Reuse):**
>
> | Q \ K  | Q_A | Q_B1 | O_A | Q_B2 | O_B |
> | ------ | --- | --- |--- | ---- | --- |
> | Q_{B1} | 0   |1    | 0   | 0    | 0   |
> | O_A    | 1   | 0    | 1   | 0    | 0   |
> | Q_{B2} | 0   |1    | 1   | 1    | 0   |
> | O_B    | 0   | 1    | 1   | 1    | 1   |
>
> The critical distinction is that CrossKV preserves $O_A$'s connection to its original context ($Q_A$), allowing Agent-B to understand $O_A$ from the same perspective as Agent-A generated it. This maintains semantic continuity while providing computational efficiency. Given that $O_A = \text{LLM}([Q_A])$ is solely a function of its immediate context $Q_A$, it follows that its attentional grounding remains there, correctly excluding $Q_{B1}$ from its attention map.
>
> Such a pattern of attention mask represents the inherent collaboration in MAS. In collaborative multi-agent systems, agents share a common goal and context. For example, in a Critic workflow setup, the Critic's prompt can be: "The Actor's response is: {Output_Actor}. Please critique it." Reusing the Actor's KV cache for Output_Actor allows the Critic to critique the output **from the same contextual perspective as the Actor**, leading to more coherent and grounded evaluations. This is not only more efficient but also **theoretically more aligned** with the goal of maintaining semantic continuity across agents.
>
> ---
> **Q2: The results are presented only with dense architectures, not with other more prevalent architectures like MoE.**
>
> **A2:** We appreciate the reviewer's comment regarding model architectures. The viability of our method depends on the components standard in decoder-only Transformers: the KV cache and the causal attention mechanism. These components are not specific to dense models but are equally foundational in MoE architectures. Since CrossKV is built upon the widely adopted prefix-caching paradigm, it is directly applicable to any autoregressive Transformer that supports KV caching, including MoE models.
>
> We have now rigorously tested this. The provided results from our experiments on Qwen2-57B-A14B-Instruct and Qwen3-30B-A3B-Instruct (see table below) serve as direct empirical validation, demonstrating that CrossKV's performance benefits seamlessly extend to the MoE setting.
>
> | Model              | Workflow          | GSM-H | GPQA  | MATH  | MMLU-Pro | MedQA | SciBench | AQUA-RAT | MedMCQA |
> | ------------------ | --------------- | ----- | ----- | ----- | -------- | ----- | -------- | -------- | ------- |
> | **Qwen2-57B-A14B** | autogen         | 9     | 15.62 | 11.6  | 20       | 18.2  | 3.41     | 44.88    | 24.6    |
> |                    | autogen+CrossKV | 35.6  | 22.9  | 31.8  | 36.4     | 45.8  | 10.62    | 54.33    | 44.2    |
> |                    | mad             | 45.6  | 28.57 | 53.2  | 40.6     | 51.8  | 14.63    | 66.54    | 52.6    |
> |                    | mad+CrossKV     | 45.4  | 26.79 | 52.6  | 40.4     | 49.8  | 16.23    | 67.32    | 47.4    |
> | **Qwen3-30B-A3B**  | autogen         | 42.4  | 45.98 | 57.8  | 61.8     | 79.8  | 28.26    | 76.77    | 68.6    |
> |                    | autogen+CrossKV | 55.8  | 50.22 | 68.8  | 67.4     | 84.6  | 27.86    | 81.89    | 71.2    |
> |                    | mad             | 56.2  | 55.58 | 93.4  | 53.6     | 54.4  | 43.09    | 81.89    | 49.6    |
> |                    | mad+CrossKV     | 59.8  | 58.26 | 93.3 | 60.4     | 61.8  | 40.08    | 80.63    | 56.4    |

---

> ### Author Response · Authors · 2025-11-20
> **Response to Reviewer Ge4H [2/2]**
>
> **Q3: It is also not clear how the semantic segments are identified in practice. The LLM identifying reusable parts of its context upon prompting seems brittle.**
>
> **A3:** We thank the reviewer for raising this important question regarding the identification of reusable segments. We sincerely appreciate the perspective offered, which has helped us recognize a potential ambiguity in our original manuscript.
>
> We wish to clarify a key point: in the current implementation, **CrossKV does not rely on the LLM itself to identify reusable segments**, which would indeed be brittle. Instead, the identification of semantic segments is handled **deterministically by the multi-agent framework**, based on the predefined interaction workflow.
>
> Specifically, the framework orchestrating the agent pipeline has explicit knowledge of when an agent’s entire output is reused as a contiguous input segment by a subsequent agent. It automatically wraps such segments (e.g., $O_A$) with special tokens: 〈reuse\_begin〉 and 〈reuse\_end〉. For example, the input to Agent-B may be structured as: [$Q_{B1}$,〈reuse\_begin〉$O_A$ 〈reuse\_end〉,$Q_{B2}$]. This tagging process is **deterministic and based solely on the workflow graph**, not on model inference, ensuring that it is both robust and reliable.
>
> We also acknowledge that the description of the two strategies in the manuscript may have caused confusion. To be clear: while CrossKV is architecturally designed to support both a deterministic tagging strategy and an LLM-prompting-based approach, **all experiments and the current implementation exclusively use the first strategy**, where segmentation is explicitly governed by the multi-agent workflow. The LLM-based approach was discussed as a more flexible, forward-looking capability but was not employed in this work due to considerations of robustness and reproducibility.
>
> We apologize for any lack of clarity in our initial writing and will revise the manuscript to unambiguously distinguish between the supported capabilities and the strategy actually used in our experiments.
>
> ---
> **Q4:In the Table 1, it is not clear why certain results will actually improve upon reusing the cached KVs. Reusing cached KVs should strictly be a inference time win, not qualitative win.**
>
> **A4:** We thank the reviewer for this insightful question. We agree that KV cache reuse is primarily an inference-time optimization; however, the observed qualitative improvements in some scenarios can be explained by the **preservation of faithful contextual representation** across collaborating agents.
>
> First, according to our response to your first question (**A1**), we show that KV cache reuse can be equivalent to computing attention under a particular, semantically meaningful mask.
>
> Moreover, as we analyzed in our introduction (Lines 63–70 in the manuscript), the KV cache is not merely a computational shortcut—it is the **compressed representation of the generative context** within the Transformer. When Agent-B reuses the KV cache of an identical segment $O_A$ generated by Agent-A, it inherits the **exact contextual understanding** that Agent-A formed during its reasoning process. In contrast, recomputing $O_A$ from scratch within Agent-B’s context forces a re-interpretation, which may subtly alter the semantic representation, especially when $O_A$ is the result of multi-step reasoning or depends on nuanced cues from earlier context.
>
> Therefore, the qualitative gains observed in Table 1 are not contradictory but stem from the fact that **KV reuse can prevent context degradation** in settings where semantic continuity is critical. It is not that reuse “adds” quality, but that it **avoids potential loss of fidelity** introduced by re-encoding the same content in a new context.
>
> ---
> **Q5:The manuscript in its current form is quite repetitive. The problem, solution, and contributions are restated in similar terms across the Abstract, Introduction, and Conclusion, diluting the paper's impact. The paper could be substantially shorter without losing any of its core technical merit.**
>
> **A5:** We appreciate the reviewer's feedback on improving the conciseness of our manuscript.
>
> The repetitive structure was a deliberate choice to emphasize the motivation and rationale behind segment-level KV cache reuse, a core concept that is foundational to our work. We sought to ensure that the reader, regardless of whether they read the Abstract, Introduction, or Conclusion, would come away with a clear understanding of the why and what of our method, as its justification is as important as its mechanism.
>
> That said, we fully agree with the reviewer that the impact can be heightened with more concise writing. We will carefully refine the entire paper, distilling the core message in each section to remove redundancy and present our ideas more powerfully. The valuable comment will undoubtedly help us improve the manuscript's overall quality.

---

> ### Author Response · Authors · 2025-11-27
> **Response to Reviewer Ge4H**
>
> We sincerely appreciate your time and expertise in reviewing our manuscript. As we have not yet received your feedback, we wanted to kindly inquire whether you might have any preliminary comments or questions that we could address at this stage. Your insights would be greatly valued in helping us further refine and strengthen our work.
>
> We truly hope that our response will contribute positively to the overall quality and impact of the paper, and we would be deeply grateful for any constructive feedback that could also support a favorable evaluation. Thank you once again for your thoughtful consideration and invaluable contribution to the peer-review process.

---

### Official Review · Reviewer_j1n9 · 2025-10-31

**Soundness:** 2
**Presentation:** 2
**Contribution:** 3
**Rating:** 4
**Confidence:** 4

**Summary:**

This paper tackles the issue of redundant computation in multi-agent LLM workflows, where agents often repeat chunks of text but prefix caching (strict reuse of KV cache based on prompt) rarely helps because prompts differ across roles and turns. The authors propose CrossKV, which shares segment-level KV cache. When a new query contains a previously generated span (i.e. continuous tokens), the system looks up that span’s hash in a "memory table" and aliases the current logical hashes to the same physical KV blocks in vLLM. This enables reuse even when prefixes don’t match. The evaluation take-away is: this content-driven reuse reduces redundant prefill/decoding across agents while remaining compatible with standard attention; the authors claim that positional (RoPE) shifts are usually negligible, and for rare long/complex cases the system caps sharing length and partially recompute to re-anchor positions.

**Strengths:**

1. Very timely and important research problem. Most prior work focuses on lossless and full KV cache reuse, e.g. prefix caching. CrossKV is a type of lossy and partial KV cache reuse, and we are already seeing this paradigm emerge in popularity recently. I believe this is an under-appreciated topic in KV cache related research that deserve more attention, given the rise of agentic workload; in agentic AI, strict prefix caching becomes less useful due to increased prompt diversity, and I appreciate the authors for motivating this emerging scenario and issue with existing cache reuse methods.

2. Solid and efficient system design. The proposed design of segment-level KV cache reuse eliminates additional I/O or data movement (e.g. duplicating KV cache blocks in GPU memory), which happens a lot in related work when positional encoding alignment is needed for partial KV cache reuse. The memory table design is efficient and scalable, as there is no need to store actual KV cache related tensors for efficient lookups. Microbenchmarks in the evaluation section are extensive and contribute to the point being made by the authors regarding efficiency.

**Weaknesses:**

1. As the authors acknowledge (e.g., in Section 3.3), misalignment in positional encoding can be a significant concern that leads to issues such as accuracy degradation; this also seems to be the main reason why CrossKV accuracy is often lower than "vanilla" in Table 1. While I agree that understanding its impact theoretically may be challenging, a more thorough empirical analysis is essential before this framework can be safely adopted in practice. Specifically, it would be helpful to examine the potential consequences when positional encodings are misaligned: What are the worst-case scenarios? Could LLMs produce irrelevant or even harmful outputs? Which types of queries or benchmarks are most sensitive to positional misalignment? Is it possible to identify when (segment-level) KV cache reuse would lead to accuracy degradation and switch to recompute instead?

2. The evaluation section of this paper lacks comparison to major baselines in the direction of lossy or partial KV cache reuse, like CacheBlend and KVShare, even though these work are mentioned in "related work".

**Questions:**

Thank you for submitting this paper to ICLR! Please refer to "weaknesses" for my questions.

---

> ### Author Response · Authors · 2025-11-20
> **Response to Reviewer j1n9 [1/2]**
>
> **Q1: Misalignment in positional encoding can be a significant concern that leads to issues such as accuracy degradation.**
>
> **A1:** We thank the reviewer for raising this critical point regarding positional encoding misalignment. To analyze the influence of potential misalignment, we implement the RoPE correction mechanism (introduced in Sec. 3.3) within the vLLM platform. The table below presents the performance of the Qwen2.5-7B-Instruct model on the AutoGen workflow across multiple benchmarks, comparing the baseline (Vanilla), standard CrossKV, and CrossKV enhanced with RoPE correction. We observe that RoPE correction can improve the performance.
> | Dataset                 | GSM-H | GPQA  | MATH | MMLU-Pro | MedQA | SciBench | AQUA-RAT | MedMCQA |
> |:----------------------- |:----- |:----- |:---- |:-------- |:----- |:-------- |:-------- |:------- |
> | Vanilla                 | 54.2  | 30.36 | 75.6 | 55.2     | 57.4  | 20.24    | 73.62    | 55      |
> | CrossKV                 | 53    | 29    | 73   | 52.8     | 57.6  | 18.44    | 74.41    | 53.6    |
> | CrossKV+RoPE correction | 52.4  | 32.14 | 71.2 | 56.2     | 56.8  | 20.04    | 75.98    | 55.6    |
>
> We also evaluated the inference efficiency, measured by Time-To-First-Token (TTFT), which reflects the latency of the prefill stage. The results below demonstrate that CrossKV (both with and without RoPE correction) maintains a significant speed advantage over the full recomputation baseline, even when reusing a substantial number of tokens.
> | \#Reused tokens | Vanilla | CrossKV | CrossKV+RoPE correction |
> | --------------- | --------------- | --------------- | ------------------------------- |
> | 0.5k            | 130.34 ms       | 92.73 ms        | 105.33 ms                       |
> | 2k              | 278.28 ms       | 113.25 ms       | 128.07 ms                       |
> | 8k              | 818.89 ms       | 114.76 ms       | 131.86 ms                                |
>
> These results confirm that our RoPE correction mechanism effectively mitigates positional misalignment issues, as evidenced by the improved scores on several benchmarks (e.g., GPQA, MMLU-Pro, AQUA-RAT), while still delivering substantial latency reductions. We will integrate these analyses and results into the revised manuscript.
>
> **Analysis about validity of segment-level KV cache sharing:** We would like to emphasize that KV cache reuse with RoPE correction is formally equivalent to computing attention with a specific, semantically consistent attention mask. This equivalence provides the theoretical foundation for our approach.
>
> Consider the following formal setup:
> - Let $Q_A$ and $O_A$ denote the input query and output of Agent-A
> - Let $Q_B = [Q_{B1}, O_A, Q_{B2}]$ denotes the input query of Agent-B, which incorporates Agent-A's output, and $O_B$ denotes the output of Agent-B.
>
> The attention mask configuration reveals the fundamental difference between the two approaches:
>
> **1. Vanilla Attention Mask (without KV Cache Reuse):**
>
> | Q \ K  | Q_A | Q_B1 | O_A | Q_B2 | O_B |
> | ------ | --- | ---- | --- | ---- | --- |
> | Q_{B1} | 0   |1    | 0   | 0    | 0   |
> | O_A    | 0   |  1    | 1   | 0    | 0   |
> | Q_{B2} | 0   |  1    | 1   | 1    | 0   |
> | O_B    | 0   | 1    | 1   | 1    | 1   |
>
> In this configuration, Agent-B must re-interpret $O_A$ without access to its original generative context ($Q_A$), potentially leading to semantic ambiguity.
>
> **2. CrossKV Attention Mask (with KV Cache Reuse):**
>
> | Q \ K  | Q_A | Q_B1 | O_A | Q_B2 | O_B |
> | ------ | --- | --- |--- | ---- | --- |
> | Q_{B1} | 0   | 1    | 0   | 0    | 0   |
> | O_A    | 1   |  0    | 1   | 0    | 0   |
> | Q_{B2} | 0   |  1    | 1   | 1    | 0   |
> | O_B    | 0   |  1    | 1   | 1    | 1   |
>
> The critical distinction is that CrossKV preserves $O_A$'s connection to its original context ($Q_A$), allowing Agent-B to understand $O_A$ from the same perspective as Agent-A generated it. This maintains semantic continuity while providing computational efficiency. Given that $O_A = \text{LLM}([Q_A])$ is solely a function of its immediate context $Q_A$, it follows that its attentional grounding remains there, correctly excluding $Q_{B1}$ from its attention map.
>
> Such a pattern of attention mask represents the inherent collaboration in MAS. In collaborative multi-agent systems, agents share a common goal and context. The reuse of $O_A$ is not an arbitrary insertion but a designed part of the workflow. For example, in a Critic workflow setup, the Critic's prompt can be: "The Actor's response is: {Output_Actor}. Please critique it." Reusing the Actor's KV cache for Output_Actor allows the Critic to understand the Actor's output from the Actor's own perspective and reasoning context. This provides a more faithful foundation for critique than re-computing the segment, which could subtly alter the contextual understanding and potentially introduce ambiguity.

---

> ### Author Response · Authors · 2025-11-20
> **Response to Reviewer j1n9 [2/2]**
>
> **Q2: It would be helpful to examine the potential consequences when positional encodings are misaligned:
> a) What are the worst-case scenarios and which types of queries are most sensitive to positional misalignment?
> b) Could LLMs produce irrelevant or even harmful outputs?
> c) Is it possible to identify when (segment-level) KV cache reuse would lead to accuracy degradation and switch to recompute instead?**
>
> **A2:** Based on our findings in **A1**, we can now provide more precise answers to your specific questions:
>
> **What are the worst-case scenarios and which types of queries are most sensitive to positional misalignment?** As detailed in Appendix Table 4, a failure scenario occurs when reusing a very long segment with complex, interdependent instructions (e.g., cross-lingual cues). In this case, positional drift can cause the model to fail to integrate the reused context with new instructions correctly. However, our results show that RoPE correction and partial recomputation effectively mitigates this issue, recovering accuracy close to the vanilla baseline.
>
> **Could LLMs produce irrelevant or even harmful outputs?** Based on our extensive evaluation and analysis in **A1**, the model does not generate irrelevant or harmful outputs due to KV cache reuse. In a Transformer, the KV cache for a sequence is the compressed representation of its context within the model. it is a deterministic function of the input tokens and model parameters, which does not hallucinate or introduce extra information.
>
> **Mitigation Strategy and Practical Guidance:** For scenarios where users prioritize maximum speed-up and can tolerate a minor accuracy trade-off, we recommend the adaptive strategy outlined in Lines 355-358: setting a maximum reuse length (e.g., N=1600 tokens). For segments longer than N, the system automatically recomputes the initial portion and reuses only the most recent N tokens. This provides a practical, configurable knob to balance performance and accuracy effectively.
>
> ---
> **Q3: The evaluation section of this paper lacks comparison to major baselines in the direction of lossy or partial KV cache reuse, like CacheBlend and KVShare, even though these work are mentioned in "related work".**
>
> **A3:** We thank the reviewer for this pertinent question regarding baseline comparisons.
>
> **Demonstrated Generality:** While our core contribution is for MAS, our method is generalizable. To directly address the reviewer's point, we conduct experiments in a RAG setting. Following the setup of KVShare, we use the Qwen2.5-7B-Instruct and evaluate on the SAMSum dataset. Notably, our reproduced vanilla baseline (without KV cache reuse) yields significantly higher Rouge-L than that reported in the original KVShare study (38.99 vs. 20.17). Even when compared against this stronger baseline, CrossKV nearly matches its performance (38.88), effectively maintaining output quality while enabling efficient KV reuse. The results confirm that our method remains effective even in RAG domain, offering a compelling performance profile.
>
> | Method           | Vanilla (KVShare) | CacheBlend@0.1 | CacheBlend@0.4 | KVShare@0.1 | KVShare@0.4 | Vanilla (Ours) | CrossKV |
> | :--------------- | :---------------- | :------------- | :------------- | :------- | :---------- | :------------- | :------ |
> | **Rouge-L**       | 20.17             | 13.92 (-6.25)          | 15.80 (-4.37)         | 15.75 (-4.42)       | 17.21 (-2.96)       | 38.99          | 38.88 (-0.11)   |
>
> **Distinctions between these methods and our CrossKV:**
>
> **1. Divergence in Problem Setting and Scope:** Works like CacheBlend and KVShare are primarily designed for RAG scenarios, which necessitate a global KV cache pool shared across many unrelated queries. In contrast, our work targets Multi-Agent Systems (MAS), where we maintain a local, session-specific KV pool dedicated to a single task's workflow. Our MAS setting is a more natural fit for exact KV reuse, as the context (e.g., an agent's output) is generated and consumed within the same session, eliminating the need for the costly upfront pre-computation (prefill) required for external RAG documents.
>
> **2. Fundamental Methodological Differences:**
> - CacheBlend introduces extra overhead for token selection and recomputation, leading to a computational complexity that is substantially higher than our lightweight, exact-match approach.
> - KVShare also introduces KV retrieval and tokens selection for recomputation. Moreover, KVShare employs a similarity-based reusing strategy, where the KV cache of a similar but not identical context is retrieved.
> - Our method is based on exact-match reuse of verbatim token sequences and has no extra computation costs for token selection. This scenario is not only highly prevalent in agent interactions but also guarantees robustness by preserving semantic integrity, as our results demonstrate.

---

> ### Author Response · Authors · 2025-11-27
> **Response to Reviewer j1n9**
>
> We sincerely appreciate your time and expertise in reviewing our manuscript. As we have not yet received your feedback, we wanted to kindly inquire whether you might have any preliminary comments or questions that we could address at this stage. Your insights would be greatly valued in helping us further refine and strengthen our work.
>
> We truly hope that our response will contribute positively to the overall quality and impact of the paper, and we would be deeply grateful for any constructive feedback that could also support a favorable evaluation. Thank you once again for your thoughtful consideration and invaluable contribution to the peer-review process.

---

> > ### Comment · Reviewer_j1n9 · 2025-11-28
> > **Thank you for the rebuttal response!**
> >
> > Thank you very much for the detailed response!
> >
> > I greatly appreciate the authors' clarification on RoPE correction and how it influences task accuracy. The TTFT reduction is quite impressive even with RoPE correction (low overhead). I also thank the authors for new results of additional baselines in the area of lossy / partial KV cache sharing. This paper presents a solid engineering solution to an important area (multi-agent system), and I believe the conference will benefit from this paper's acceptance, thus raising my score from 4 to 6.
> >
> > **More thoughts:** This paper could be even stronger with a more solid explanation and implementation of the "semantic segment" mechanism. Currently, the authors argue that segment-level KV cache sharing is a better and smarter paradigm, while the semantic segments are in fact determined in a manual and deterministic way (using the pre-defined brackets as boundaries).

---

> > > ### Comment · Reviewer_j1n9 · 2025-11-28
> > > **Bug with OpenReview; Score will be raised after bug fix**
> > >
> > > OpenReview seems to be suffering from a bug that disables reviewers from editing their reviews and scores. The mentioned score raise above will be made once the system is back online.

---

> > > ### Author Response · Authors · 2025-11-28
> > > **Response to Reviewer j1n9**
> > >
> > > We sincerely thank the reviewer for the encouraging feedback and for raising the score to a 6. We are also grateful for the constructive suggestion regarding the "semantic segment" mechanism. We agree that a more flexible approach would further strengthen this part. We will take this insight into account and refine our explanation of the mechanism, exploring ways to move beyond manually defined boundaries.

---

### Official Review · Reviewer_V51C · 2025-11-01

**Soundness:** 3
**Presentation:** 2
**Contribution:** 2
**Rating:** 4
**Confidence:** 3

**Summary:**

This paper introduces CrossKV, a segment-level KV cache sharing mechanism that enables flexible reuse of intermediate computations across agents in multi-agent workflows without requiring prefix alignment. Built on vLLM, it decouples cache reuse from rigid prefix matching, allowing agents to share semantic segments at arbitrary positions.

**Strengths:**

1. The segment-level sharing mechanism effectively overcomes the limitations of prefix-based caching in multi-agent systems.
2. The implementation demonstrates practical system-level gains, with notable improvements in both throughput and task performance.

**Weaknesses:**

1. Section 3.1.2 directly presents the attention map visualization comparing cases with and without segment-level sharing in Figure 1. However, at this point, the mechanism of segment-level KV sharing remains unclear—under what conditions (e.g., a certain degree of token similarity) does sharing occur, and what are the typical patterns of such segments? Section 3.1.1 also lacks rigorous and interpretable formulations, relying solely on textual descriptions, which makes Figure 1 difficult to understand.
2. It is still unclear how CrossKV is integrated into existing multi-agent frameworks. Does it replace the original natural language communication among agents, or do agents still communicate via natural language while additionally performing KV sharing?
3. CrossKV seems inapplicable to heterogeneous LLM-based multi-agent systems, as it does not address potential discrepancies in hidden state dimensions or distributions across different backbone LLMs. Restricting the method to homogeneous MASs substantially limits its contribution.
4. How are the `<reuse begin>` and `<reuse end>` tags obtained? If they are generated by the model itself, how is the correctness or reasonableness of their positions ensured?
5. When these KV caches are directly reused, does this lead to semantic discontinuity? Directly embedding the cached segments from one agent into another agent’s input without adaptation seems problematic.

**Questions:**

See Weakness

---

> ### Author Response · Authors · 2025-11-20
> **Response to Reviewer V51C [1/2]**
>
> **Q1:
> a) The mechanism of segment-level KV sharing remains unclear—under what conditions (e.g., a certain degree of token similarity) does sharing occur, and what are the typical patterns of such segments?
> b) Section 3.1.1 also lacks rigorous and interpretable formulations, relying solely on textual descriptions, which makes Figure 1 difficult to understand.**
>
> **A1:** The mechanism of segment-level KV sharing is triggered by a direct content overlap between agent outputs and inputs. Below, we formalize the conditions and patterns to clarify the methodology.
>
>  **Pattern for Sharing:** Sharing is applicable when a token sequence generated by one agent is *verbatim* incorporated as a contiguous segment into another agent's input prompt. This is prevalent in pipelined agent workflows.
> Formally, Let $O_A = \text{LLM}([Q_A])$ be the output of Agent-A where $Q_A$ denotes the input query of Agent-A. For Agent-B, let its input be $Q_B = [Q_{B1}, O_A, Q_{B2}]$. During the pre-filling phase for $Q_B$, the KV cache for the exact segment $O_A$, computed during Agent-A's decoding, is reused. This constitutes the "segment-level" sharing pattern.
>
> **Interpretation of Figure 1:** Suppose $O_B = \text{LLM}([Q_B])$. In this context, Figure 1 demonstrates the attention map between $O_B$ and $Q_B=[Q_{B1}, O_A, Q_{B2}]$. The segment between the red lines corresponds to the attention map between $O_B$ and $O_A$. The visualization shows that the attention pattern of reusing $O_A$'s KV cache is similar to the pattern of vanilla attention without segment-level KV sharing, confirming that the reuse of identical KV cache preserves semantic integrity without introducing noise or leakage. An example of this $O_A$-in-$Q_B$ pattern is illustrated in Table 2 of the appendix.
>
> ---
> **Q2: It is still unclear how CrossKV is integrated into existing multi-agent frameworks. Does it replace the original natural language communication among agents, or do agents still communicate via natural language while additionally performing KV sharing?**
>
> **A2:** Thank you for this question. CrossKV does **not** replace natural language communication among agents; it is a complementary performance optimization. Built upon vLLM, our CrossKV can reuse the already-computed KV cache for identical text segments across these interactions.
>
> **Communication Paradigm:** Agents fundamentally still communicate via natural language. The content of one agent's output (e.g., $O_A$ from Agent-A) is passed as a natural language string to the next agent (Agent-B). CrossKV does not alter this semantic-level interaction.
>
> **Optimization Mechanism:** CrossKV's core reuse mechanism is built atop the vLLM platform. Its operation is as follows: if a segment in an agent's input is delimited by 〈reuse\_begin〉/〈reuse\_end〉 tags and the enclosed content is a verbatim copy of a previously generated sequence (like $O_A$), then the corresponding KV cache is retrieved and reused. This non-intrusive optimization, detailed in Sec. 3.2.1 of our manuscript, saves substantial computation while ensuring semantic integrity.
>
> **How is it integrated?** The integration is lightweight and non-invasive. Built upon vLLM, our method is largely independent of the higher-level agent framework. The framework's only responsibility is to wrap any text segment that is known to be a repeat of a prior agent's output with a pair of special tags. For instance, if Agent-B's input reuses Agent-A's output $O_A$, the input would be structured as [$Q_{B1}$, 〈reuse\_begin〉$O_A$〈reuse\_end〉, $Q_{B2}$]. Our inference engine encounters these tags, it triggers the internal KV cache reuse logic. Since the agent workflow defines which outputs are routed to which inputs, this tagging process can be fully automated, requiring minimal changes to the existing framework code.
>
> ---
> **Q3: CrossKV seems inapplicable to heterogeneous LLM-based multi-agent systems, as it does not address potential discrepancies in hidden state dimensions or distributions across different backbone LLMs. Restricting the method to homogeneous MASs substantially limits its contribution.**
>
> **A3:** We appreciate the reviewer's point regarding heterogeneous systems. Our work specifically targets and optimizes the homogeneous MAS setting, which is both common and highly practical for building complex agent systems. It is also worth noting that in heterogeneous systems, individual LLMs can still employ CrossKV to recycle their own KV cache segments for internal reuse.
>
> Extending CrossKV to heterogeneous systems is a valuable but separate research path, as it would require solving the challenge of cross-model state mapping, a problem orthogonal to the segment-level reuse mechanism we propose. We believe our work provides a strong foundation, and we will explicitly discuss this exciting direction for future work.

---

> ### Author Response · Authors · 2025-11-20
> **Response to Reviewer V51C [2/2]**
>
> **Q4: How are the \<reuse\_begin\> and \<reuse\_end\> tags obtained? If they are generated by the model itself, how is the correctness or reasonableness of their positions ensured?**
>
> **A4:** We sincerely appreciate the perspective offered, which has helped us recognize a potential ambiguity in our original manuscript. We wish to clarify a key point: in the current implementation, **CrossKV does not rely on the LLM itself to identify reusable segments**. Instead, the identification of semantic segments is handled **deterministically by the multi-agent framework**, based on the predefined interaction workflow.
>
> Specifically, when the workflow dictates that Agent-B will reuse Agent-A's output $O_A$, the framework injects the tags around $O_A$ when constructing Agent-B's input prompt. Hence, the input of Agent-B would be structured as [$Q_{B1}$, 〈reuse\_begin〉$O_A$〈reuse\_end〉, $Q_{B2}$], where $Q_{B1}$ and $Q_{B2}$ denotes the instruction prompts for Agent-B. This tagging process is **deterministic and based solely on the workflow graph**, not on model inference, ensuring that it is both robust and reliable. For more on integration, please refer to our response in A2.
>
> We also acknowledge that the description of the two strategies in the manuscript may have caused confusion. To be clear: while CrossKV is architecturally designed to support both a deterministic tagging strategy and an LLM-prompting-based approach, **all experiments and the current implementation exclusively use the first strategy**, where segmentation is explicitly governed by the multi-agent workflow. The LLM-based approach was discussed as a more flexible, forward-looking capability but was not employed in this work due to considerations of robustness and reproducibility.
>
> We apologize for any lack of clarity in our initial writing and will revise the manuscript to unambiguously distinguish between the supported capabilities and the strategy actually used in our experiments.
>
> ---
> **Q5: When these KV caches are directly reused, does this lead to semantic discontinuity? Directly embedding the cached segments from one agent into another agent’s input without adaptation seems problematic.**
>
> **A5:** We thank the reviewer for raising this crucial point. We argue that direct KV cache reuse does not cause semantic discontinuity but rather preserves semantic fidelity and enhances contextual consistency.
> We would like to emphasize that KV cache reuse with RoPE correction is formally equivalent to computing attention with a specific, semantically consistent attention mask.
>
> Consider the following formal setup:
> - Let $Q_A$ and $O_A$ denote the input query and output of Agent-A
> - Let $Q_B = [Q_{B1}, O_A, Q_{B2}]$ denotes the input query of Agent-B, which incorporates Agent-A's output, and $O_B$ denotes the output of Agent-B.
>
> The attention mask configuration reveals the fundamental difference between the two approaches:
>
> **1. Vanilla Attention Mask (without KV Cache Reuse):**
> | Q \ K  | Q_A |Q_B1 | O_A | Q_B2 | O_B |
> | ------ | --- | ---- | --- | ---- | --- |
> | Q_{B1} | 0   | 1    | 0   | 0    | 0   |
> | O_A    | 0   | 1    | 1   | 0    | 0   |
> | Q_{B2} | 0   | 1    | 1   | 1    | 0   |
> | O_B    | 0   | 1    | 1   | 1    | 1   |
>
> In this configuration, Agent-B must re-interpret $O_A$ without access to its original generative context ($Q_A$), potentially leading to semantic ambiguity.
> **2. CrossKV Attention Mask (with KV Cache Reuse):**
>
> | Q \ K  | Q_A | Q_B1 | O_A | Q_B2 | O_B |
> | ------ | --- | ---- | --- | ---- | --- |
> | Q_{B1} | 0   | 1    | 0   | 0    | 0   |
> | O_A    | 1   | 0    | 1   | 0    | 0   |
> | Q_{B2} | 0   | 1    | 1   | 1    | 0   |
> | O_B    | 0   | 1    | 1   | 1    | 1   |
>
> The critical distinction is that CrossKV preserves $O_A$'s connection to its original context ($Q_A$), allowing Agent-B to understand $O_A$ from the same perspective as Agent-A generated it. This maintains semantic continuity while providing computational efficiency. Given that $O_A = \text{LLM}([Q_A])$ is solely a function of its immediate context $Q_A$, it follows that its attentional grounding remains there, correctly excluding $Q_{B1}$ from its attention map.
>
> Such a pattern of attention mask represents the inherent collaboration in MAS. In collaborative multi-agent systems, agents share a common goal and context. The reuse of $O_A$ is not an arbitrary insertion but a designed part of the workflow. For example, in a Critic workflow setup, the Critic's prompt can be: "The Actor's response is: {Output_Actor}. Please critique it." Reusing the Actor's KV cache for Output_Actor allows the Critic to understand the Actor's output from the Actor's own perspective and reasoning context. This provides a more faithful foundation for critique than re-computing the segment, since recomputing could subtly alter the contextual understanding and potentially introduce ambiguity.

---

> ### Author Response · Authors · 2025-11-27
> **Response to Reviewer V51C**
>
> We sincerely appreciate your time and expertise in reviewing our manuscript. As we have not yet received your feedback, we wanted to kindly inquire whether you might have any preliminary comments or questions that we could address at this stage. Your insights would be greatly valued in helping us further refine and strengthen our work.
>
> We truly hope that our response will contribute positively to the overall quality and impact of the paper, and we would be deeply grateful for any constructive feedback that could also support a favorable evaluation. Thank you once again for your thoughtful consideration and invaluable contribution to the peer-review process.

---

### Author Response · Authors · 2025-11-24
**General Response**

Dear Area Chair and Reviewers,

We sincerely thank all the reviewers for their insightful and constructive feedback, which has been invaluable in improving our manuscript. In response to the comments and questions raised, we have thoroughly revised the paper to address each point in detail. The key enhancements are summarized as follows:

**Clarification of the Mechanism (Addressing Reviewers V51C & Ge4H):** We have formally defined the segment-level KV sharing mechanism, clarifying that it is activated by the verbatim reuse of an agent's output as a contiguous segment in another agent’s input—a common pattern in pipelined agent workflows. The deterministic use of special tags for segment identification within the agent framework has also been explicitly detailed to enhance clarity.

**Theoretical Justification (Addressing Reviewers j1n9, Ge4H & taJt):** To address concerns regarding semantic stability, we have introduced a formal equivalence proof showing that KV cache reuse with RoPE correction is equivalent to computing attention under a specific, semantically consistent mask. This analysis reinforces that our method preserves contextual integrity across collaborating agents.

**Expanded Experimental Validation (Addressing Reviewers j1n9, Ge4H & taJt):** We conducted extensive new experiments:

- RAG Comparison: Evaluation on RAG benchmarks confirms that CrossKV performs competitively against a strong vanilla baseline and maintains robustness, whereas methods such as CacheBlend and KVShare exhibit notable performance degradation.

- MoE Architectures: Validation on large-scale MoE models (e.g., Qwen2-57B-A14B and Qwen3-30B-A3B) demonstrates that CrossKV consistently preserves model accuracy while improving inference efficiency.

- Complex Reasoning: Experiments on the challenging GAIA benchmark with a 72B model show that CrossKV not only accelerates inference but also enhances task accuracy in complex, multi-step reasoning settings.


We believe these revisions and additions have substantially strengthened the manuscript and fully address the reviewers' concerns. We deeply appreciate the reviewers' time and expertise and remain open to any further questions or suggestions.

&nbsp;

Sincerely,

The Authors

---

### Meta-Review · Area_Chair_wiNc · 2026-01-13

**Summary:**

The paper proposes CrossKV, a system designed to optimize inference in Large Language Model (LLM) based multi-agent systems (MAS). It addresses a specific inefficiency where agents frequently exchange identical text segments (e.g., an agent's output becoming another's input), yet standard prefix caching fails because the prompt prefixes differ. The authors introduce a segment-level KV cache sharing mechanism built on vLLM, allowing exact-match segments to be reused regardless of their position.

**Reviewer Concerns:**

* Positional Encoding (RoPE) Issues: Reviewers j1n9 and taJt questioned the impact of position shifts on accuracy. The authors implemented a RoPE correction mechanism and adaptive partial recomputation, showing that they could recover accuracy losses with minimal latency overhead].

* Baselines: Reviewers requested comparisons to other caching methods like CacheBlend and KVShare. The authors provided new experiments showing CrossKV outperforms these methods in the exact-match scenarios typical of MAS].

* Generalizability: In response to Reviewer Ge4H's concern about dense-only evaluations, the authors provided new results for Mixture-of-Experts (MoE) models (Qwen2-57B-A14B), proving the method's applicability to modern architectures].

* the theoretical justification for why reuse preserves semantic integrity is largely empirical (visual attention maps).

* The authors noted that for extremely long segments or complex cross-lingual instructions, simple RoPE correction mig.ht still lead to minor accuracy degradation due to attention "drift." To mitigate this, they introduced an Adaptive Recomputation Strategy

**Reviewer Scores:**

Given above, the edits made during rebuttal are drastic (RoPE issue) and the are some remaining issue (see the theoretic justification part) therefore, not all reviewers would have changed their score to acceptance.

---

### Decision · Program_Chairs · 2026-01-26

Reject